# Design of novel cyanovirin-N variants by modulation of binding dynamics through distal mutations

I Can Kazan[1,2†], Prerna Sharma[2†], Mohammad Imtiazur Rahman[2†], Andrey Bobkov[3], Raimund Fromme[2], Giovanna Ghirlanda[2*], S Banu Ozkan[1*]

[1]Center for Biological Physics and Department of Physics, Arizona State University, Tempe, United States; [2]School of Molecular Sciences, Arizona State University, Tempe, United States; [3]Sanford Burnham Prebys Medical Discovery Institute, La Jolla, United States

**Abstract** We develop integrated co-evolution and dynamic coupling (ICDC) approach to identify, mutate, and assess distal sites to modulate function. We validate the approach first by analyzing the existing mutational fitness data of TEM-1 β-lactamase and show that allosteric positions co-evolved and dynamically coupled with the active site significantly modulate function. We further apply ICDC approach to identify positions and their mutations that can modulate binding affinity in a lectin, cyanovirin-N (CV-N), that selectively binds to dimannose, and predict binding energies of its variants through Adaptive BP-Dock. Computational and experimental analyses reveal that binding enhancing mutants identified by ICDC impact the dynamics of the binding pocket, and show that rigidification of the binding residues compensates for the entropic cost of binding. This work suggests a mechanism by which distal mutations modulate function through dynamic allostery and provides a blueprint to identify candidates for mutagenesis in order to optimize protein function.

**\*For correspondence:**
Giovanna.Ghirlanda@asu.edu (GG);
banu.ozkan@asu.edu (SBO)

†These authors contributed equally to this work

**Competing interest:** The authors declare that no competing interests exist.

## Editor's evaluation

A computational approach is proposed to identify mutations in enzymes that might impact their interactions with substrates. For one enzyme, in particular, the predictions are validated through experiments, using multiple techniques. Taken together, these data lead to non-trivial conclusions in regard to the nature of allosteric effects, albeit it remains unclear whether these conclusions will apply more broadly when other enzymes are examined.

## Introduction

The evolutionary history of a protein comprises the ensemble of mutations acquired during the course of its evolutionary trajectory across different species, and contains valuable information on which residue positions contribute the most to a given protein's 3D-fold and function based on their conservation (*Campbell et al., 2016*; *Rivoire et al., 2016*; *Yang et al., 2016*). Furthermore, the subset of positions that are co-evolved (i.e., correlated mutational sites) provide clues on specific, native-state interactions. Pairwise residue contacts inferred from co-evolved positions within a protein family can be used as distance restraints to accurately model 3D structures (*de Juan et al., 2013*; *Hopf et al., 2019*; *Kamisetty et al., 2013*; *Kim et al., 2014*; *Tripathi et al., 2015*). Recent revolutionary successes in accurate predictions of 3D protein structures combine these methods with machine learning strategies, that is, deep learning (*Jumper et al., 2021*; *Wang et al., 2016*; *Xu, 2019*). Co-evolved positions also embed information on protein function, for example, revealing how factors such as binding affinity

and specificity are modulated across evolutionary history and species (*Rivoire et al., 2016*; *Salinas and Ranganathan, 2018*; *Torgeson et al., 2022*). However, accessing, interpreting, and applying this information in a predictive manner is very challenging; mutations observed in the evolutionary history are often distal from the functional sites, implying that protein dynamics are responsible for their effects on function and that these sites act as distal allosteric regulators of function (*Campitelli et al., 2020a*; *Modi et al., 2021a*; *Romero and Arnold, 2009*; *Salinas and Ranganathan, 2018*; *Tokuriki et al., 2012*; *Torgeson et al., 2022*; *Wei et al., 2016*).

Molecular dynamics (MD) simulations can capture protein dynamics and reveal the impact of distal mutations on function (*Bowman and Geissler, 2012*; *Campbell et al., 2016*; *Campitelli et al., 2020a*; *Jiménez-Osés et al., 2014*; *Kolbaba-Kartchner et al., 2021*; *Modi et al., 2021a*; *Yang et al., 2016*). However, the computational cost of MD simulations of sufficient length can be prohibitively high; further, it's often far from straightforward to forge a clear connection to function. To bridge this gap, we developed a framework to quickly evaluate MD trajectories and identify the sensitivity of a given position to mutation based on its intrinsic flexibility, which we assess using our dynamic flexibility index (DFI) metric, and on its dynamic coupling with functionally critical positions assessed by dynamic coupling index (DCI) (*Campitelli et al., 2018*; *Gerek and Ozkan, 2011*; *Kumar et al., 2015b*; *Larrimore et al., 2017*). DFI measures the resilience of a position by computing the total fluctuation response and thus captures the flexibility/rigidity of a given position. Applying DFI to several systems, we showed that rigid positions such as hinge sites contribute the most to equilibrium dynamics, and that mutations at hinge sites significantly impact function regardless of the distance from active sites (*Kim et al., 2015*; *Kolbaba-Kartchner et al., 2021*; *Modi et al., 2021b*, *Modi et al., 2018*; *Modi and Ozkan, 2018*; *Zou et al., 2021*; *Zou et al., 2015*). DCI measures the dynamic coupling between residue pairs and thus identifies positions most strongly coupled to active/binding sites; these positions point to possible allosteric regulation sites important for modulating function in adaptation and evolution (*Butler et al., 2015*; *Modi et al., 2021a*, *Campitelli et al., 2021*; *Kuriyan and Eisenberg, 2007*; *Lu and Liang, 2009*; *Modi and Ozkan, 2018*; *Ose et al., 2020*; *Risso et al., 2018*; *Wodak et al., 2019*).

In this paper, we present integrated co-evolution and dynamic coupling (ICDC) approach to identify distal allosteric sites, and to assess and predict the effects of mutations on these sites on function. We propose a system to classify residue positions in a binary fashion based on co-evolution (co-evolved, 1 or not, 0) and dynamic coupling by DFI and DCI (dynamically coupled 1, or not, 0) with the functionally critical sites. This classification captures the complementarity of dynamics-based and sequence-based methods. We hypothesize that positions belonging to category (**1,1**), that is, positions both co-evolved and dynamically coupled with the functional sites, will have the largest effect on function.

We validate our hypothesis first by analyzing the existing mutational fitness data for TEM-1 β-lact-amase, available for every position of the protein (*Stiffler et al., 2015*). In agreement with our hypothesis, we find that mutations on category (**1,1**) positions significantly modulate the function. A large fraction of mutations enhancing enzymatic activity correspond to category (**1,1**) irrespective of distance from the active site. Second, we apply our ICDC approach to blindly predict and experimentally validate mutations that allosterically modulate dimannose binding in a natural lectin, cyanovirin-N (CV-N). CV-N binds dimannose with nanomolar affinity and remarkable specificity (*Barrientos et al., 2003*; *Botos and Wlodawer, 2005*; *Botos and Wlodawer, 2003*; *Mori and Boyd, 2001*; *O'Keefe et al., 2003*). It is part of the CV-N family, found in a wide range of organisms including cyanobacterium, ascomycetous fungi, and fern (*Koharudin et al., 2008*; *Koharudin and Gronenborn, 2013*; *Patsalo et al., 2011*; *Percudani et al., 2005*; *Qi et al., 2009*). While the 3D folds is remarkably conserved in all experimentally characterized members, the affinity and specificity for different glycans and, in particular, to dimannose varies significantly (*Koharudin et al., 2009*; *Koharudin et al., 2008*; *Matei et al., 2016*; *Woodrum et al., 2013*). To design CV-N variants with improved binding affinities for dimannose based on distal allosteric coupling, we binned each position in one of the four categories based on computed DFI, DCI, and co-evolution rates. We explored mutations at these sites based on frequency in the sequence alignment. After obtaining the mutant models through MD simulations, we assessed the impact of each naturally observed mutation on binding affinity by docking dimannose to the mutant models via Adaptive BP-Dock (*Bolia et al., 2014a*; *Bolia et al., 2014b*; *Bolia and Ozkan, 2016*). We chose position I34, which belongs to category (**1,1**) and is 16 Å away from the binding pocket, for experimental validation. We found that mutations I34K/L/Y had a diverse effect on glycan

binding, either improving by twofold or abolishing completely. Through experimental and MD studies we show that the observed improvement in binding affinity is due to changes in the dynamics of residues in the binding pocket; mutation I34Y leads to rigidification of binding sites, thus compensating the entropic cost of binding (*Breiten et al., 2013*; *Chodera and Mobley, 2013*; *Cornish-Bowden, 2002*; *Fox et al., 2018*). Mutations at an additional position (A71T/S) from category (**1,1**) showed evidence of the same allosteric mechanism governing the modulation of binding dynamics. Overall, this study provides not only a new approach to identify distal sites whose mutations modulate binding affinity, but also sheds light into mechanistic insights on how distal mutations modulate binding affinity through dynamics allostery.

## Results and discussion
### Combining long-range dynamic coupling analysis with co-evolution allows to identify distal sites that contribute to functional activity

With our ICDC approach, we aim to explore the role of dynamics versus evolutionary coupling (EC) as well as the role of rigidity versus flexibility in allosterically modulating active/binding site dynamics. To this extent, we created four unique categories that classify residue positions based on residue DFI score, DCI score, and co-evolutionary score: category (**1,1**) is dynamically and co-evolutionarily coupled rigid sites (exhibiting %DFI values 0.2 or lower, showing 0.7 or higher %DCI with the binding site, and showing 0.6 or higher co-evolution scores with the binding site); category (**1,0**) is dynamically coupled but co-evolutionarily not coupled sites; category (**0,1**) is dynamically not coupled but co-evolutionarily coupled sites; category (**0,0**) is dynamically not coupled, and co-evolutionarily not coupled flexible sites (exhibiting %DFI values 0.7 or higher) (*Supplementary file 1* and *Supplementary file 2*; ); importantly, this classification is based on two independent statistical approaches thus compensate the noise of individual approaches. Based on our evolutionary analysis (*Campitelli et al., 2020a*; *Modi et al., 2021b*; *Modi and Ozkan, 2018*), we hypothesize that category (**1,1**) would impact protein activity or binding affinity the most.

To test our hypothesis, we first analyzed the deep mutational scanning data available for the TEM-1 β-lactamase, correlating changes in ampicillin degradation activity (e.g., MIC values) with mutations to all possible amino acids at each position (*Stiffler et al., 2015*). The experimental results showed that amino acid substitutions at the catalytic site residues of TEM-1 negatively impacted activity. Mutations at other positions also affected activity; while most mutations were deleterious, surprisingly, others resulted in increased activity. The impact of mutations on dynamics and function of TEM-1 have been heavily explored but the distal mutational effects are still poorly understood (*Kolbaba-Kartchner et al., 2021*; *Modi et al., 2021b*; *Modi and Ozkan, 2018*; *Salverda et al., 2010*; *Schneider et al., 2021*; *Stiffler et al., 2015*; *Thomas et al., 2010*; *Zimmerman et al., 2017*; *Zou et al., 2015*). We applied our approach by obtaining DFI, DCI, and co-evolution scores for every position of TEM-1 and binning residue positions into each ICDC category (*Supplementary file 1* and *Supplementary file 5*). We constructed fitness distributions for each category using the experimentally measured single mutant relative fitness values for all mutations per position provided in the dataset (*Figure 1*).

We found that category (**1,1**) positions show the highest impact, both significantly enhancing and reducing ampicillin degradation by TEM-1 (*Figure 1A&C*). In addition, category (**0,0**) residue mutations (i.e., the exact opposite of category (**1,1**)) lie within the neutral-like activity range defined by *Stiffler et al., 2015*, suggesting that mutations on positions that neither co-evolve nor dynamically couple to active site do not affect the function significantly. Category (**1,0**) residues enhance activity more than those in the neutral category (**0,0**). Mutations in category (**0,1**) positions also modulate function in both positive and negative direction, albeit not as strongly as those in category (**1,1**). However, mutations that negatively impact activity are conspicuously under-represented in the multiple sequence alignment (MSA) of native sequences (*Figure 1B*), particularly in category (**1,1**). This finding implies nature mostly allows mutations that don't compromise fold and function: Negative selection (i.e., elimination of amino acid types that are detrimental to the folding) is a major force in shaping the mutational landscape (*Jana et al., 2014*; *Modi et al., 2021a*; *Morcos, 2020*; *Morcos et al., 2014*; *Morcos et al., 2013*). Thus, the use of conservation information from MSA is a useful tool in eliminating deleterious amino acid substitutions in protein design.

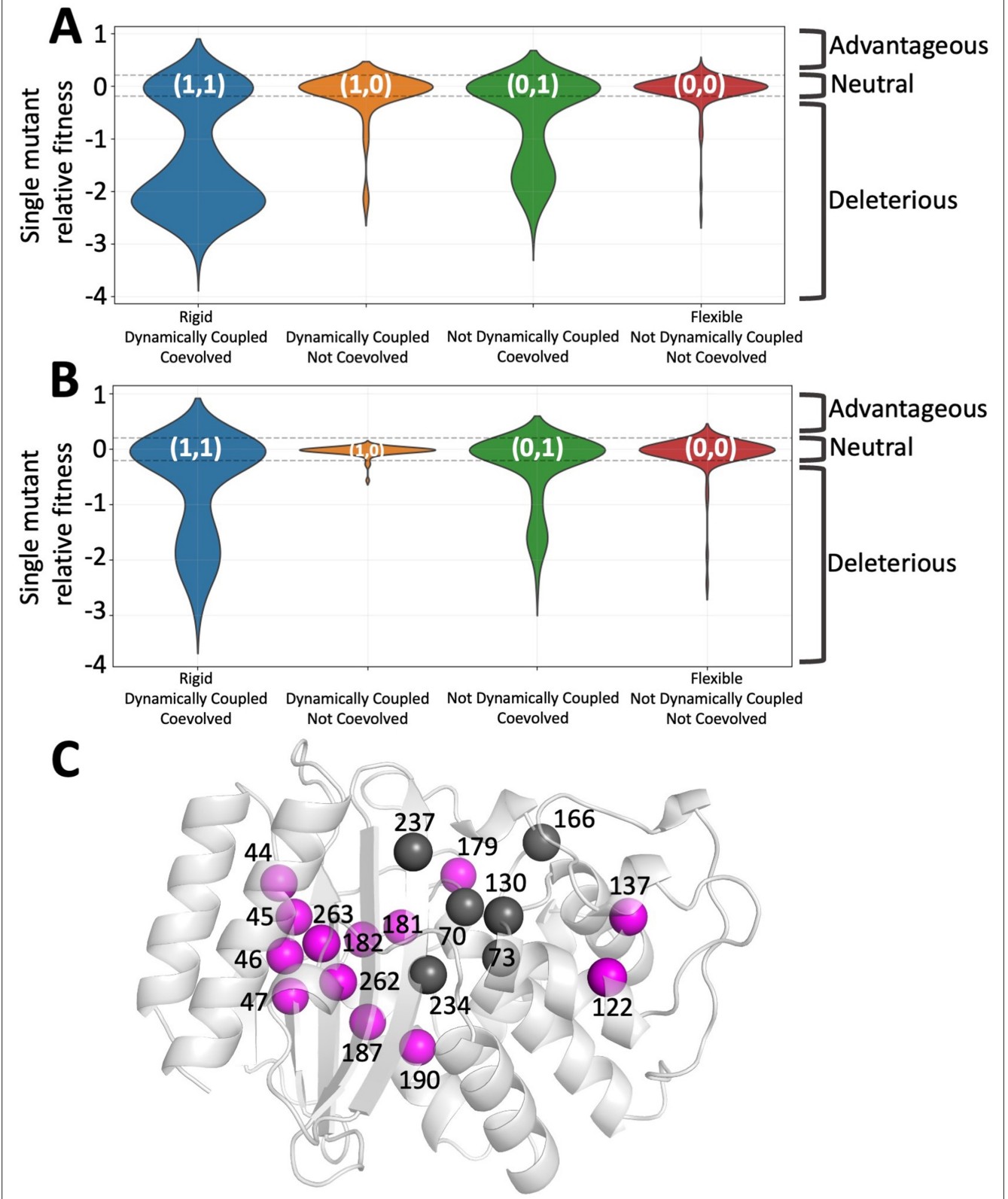

**Figure 1.** Integrated co-evolution and dynamic coupling (ICDC) categories based on the dynamics and co-evolutionary analyses applied on TEM-1 β-lactamase. (**A**) The distributions in the form of violin plots are obtained for each ICDC category using all available experimental mutational data (*Stiffler et al., 2015*). (**B**) Violin plots showing the fitness values for amino acid substitutions observed in the natural sequences. (**C**) The category (**1**,**1**) positions are mapped on 3D structure. The catalytic site residues are shown in dark gray whereas category (**1**,**1**) positions are shown in magenta color. The function altering category (**1**,**1**) positions are widely distributed over the 3D structure.

Our ICDC selection criteria effectively identifies residue positions and their amino acid substitutions that could fine-tune function without leading to a functional loss; and category (**1,1**) residues have the largest impact on function irrespective of their distance from active site (*Figure 1C*).

## Application of ICDC approach to modulate CV-N binding affinity through distal mutations

CV-N is a small (11 kDa) natural lectin isolated from cyanobacterium *Nostoc ellipsosporum* which comprises two quasi-symmetric domains, A (residues 1–38/90–101) and B (residues 39–89 respectively), that are connected to each other by a short helical linker. Despite almost having identical structures, the domains show relatively low sequence homology (28% sequence identity and 52% similarity). Functionally, they both bind dimannose, yet the affinity is quite different, with domain B having tighter binding affinity ($K_d = 15.3$ μM), and domain A showing weak affinity ($K_d = 400$ μM) (*Balzarini, 2007*; *Bolmstedt et al., 2001*; *Li et al., 2015*).

To simplify our analyses, we used a designed CV-N variant, P51G-m4, that contains a single high-affinity dimannose binding site (domain B), folds exclusively as a monomer in physiological conditions, and is more stable to thermal denaturation than wild type (*Fromme et al., 2008*; *Fromme et al., 2007*). The binding pocket of domain B of CV-N has been subjected to intense scrutiny to glean information on the origin of its binding specificity for dimannose (*Bewley, 2001*; *Bolia et al., 2014b*; *Botos and Wlodawer, 2003*; *Li et al., 2015*; *Vorontsov and Miyashita, 2009*). Previous mutational studies on the binding pocket residues have shown their importance in modulating interaction with dimannose (*Barrientos et al., 2006*; *Bolia et al., 2014b*; *Chang and Bewley, 2002*; *Matei et al., 2008*). All known substitutions of the binding residues led to decreased binding affinity for dimannose on domain B (*Bolia et al., 2014b*; *Fujimoto and Green, 2012*; *Kelley et al., 2002*; *Matei et al., 2011*; *Ramadugu et al., 2014*). Evolutionary analyses shows that the majority of the binding site residues are conserved in CV-N glycan interactions, suggesting that affinity is already optimized at the binding site (*Koharudin et al., 2008*; *Percudani et al., 2005*). We hypothesized that amino acid substitutions at distal positions could enhance the dimannose affinity of CV-N by rigidification of the binding site and applied our ICDC approach to CV-N to identify positions in each category (*Supplementary file 2*).

We generated models of CV-N variants in each ICDC category by mutating these positions to amino acid types observed in the MSA of CV-N family members, choosing the subset of sequences that have binding sites with identical or similar amino acid composition to P51G-m4 CV-N. As discussed above, this approach allows us to identify amino acid substitutions with the least impact on fold. All the substitutions identified (104 variants in total) were modeled using the crystal structure of P51G-m4 CV-N (*Fromme et al., 2008*) and subjected to MD simulations (*Abraham et al., 2015*; *Van Der Spoel et al., 2005*). The best conformation sampled for each variant obtained from

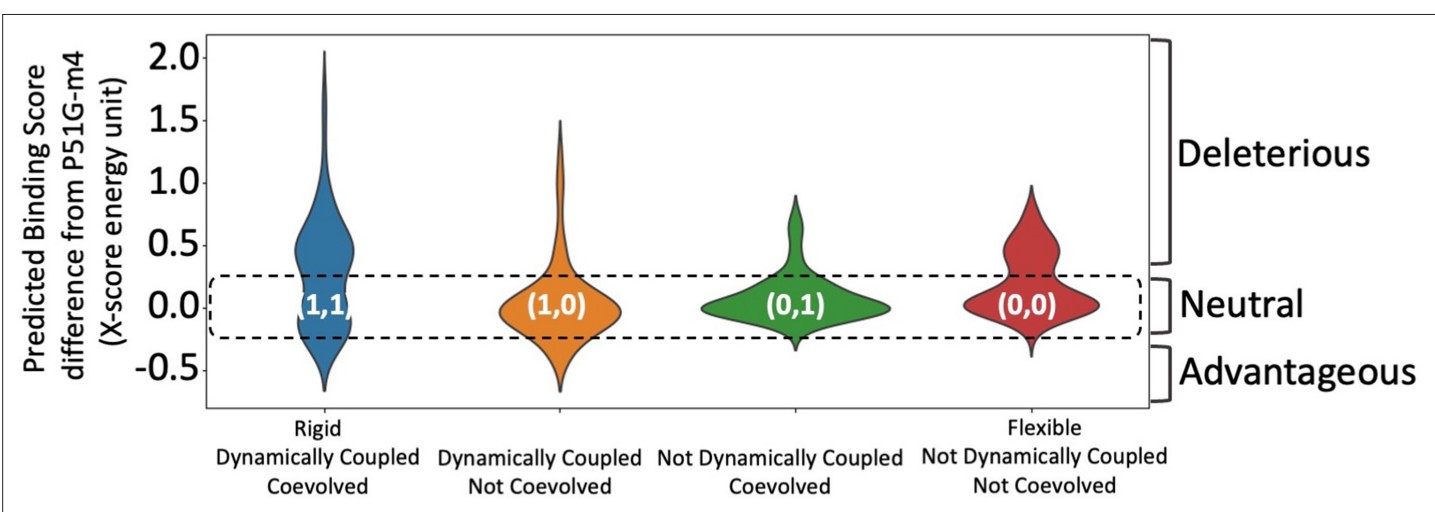

**Figure 2.** Predicted binding energies for each integrated co-evolution and dynamic coupling (ICDC) category. Mutations in category (**1,1**) positions comprise the highest number of binding energy enhancing mutations as well as deleterious mutations. Mutations in category (**0,0**) positions are mostly near neutral (category (**1,1**) and (**0,0**) p value <0.3).

equilibrated production trajectories was used as a model for dimannose docking analysis. We evaluated the variants using Adaptive BP-Dock (*Bolia and Ozkan, 2016*), a computational docking tool that incorporates both ligand and receptor flexibility to accurately sample binding-induced conformations, and ranks them using X-scores binding energy units (XEUs) (*Figure 4—figure supplement 1*). In previous work on CV-N this method yielded good correlations with experimentally measured binding affinities ($K_d$), and established –6.0 XEU as a good threshold to differentiate variants that bind dimannose from 'non-binders' (*Bolia et al., 2014b*; *Li et al., 2015*; *Woodrum et al., 2013*). Here, we applied Adaptive BP-Dock initially on wild-type CV-N and its variants, P51G-m4 and mutDB (a mutant in which binding by domain B has been obliterated) and the results recapitulate the success of previous studies (*Supplementary file 3*). This result shows that Adaptive BP-Dock can correctly assess the dimannose binding of CV-N and its variants, thus, we applied it on new P51G-m4 CV-N variants to predict the impact of mutations on dimannose binding. *Figure 2* shows the distribution of changes in predicted binding energy scores relative to the P51G-m4 energy scores for mutations belonging to each binary category: a positive change in binding score represents an unfavorable effect on binding, and, conversely, a negative change in the score indicates an enhancement in binding.

The substitutions on positions in category (**1,1**) (*Figure 2*) yield a wide range of change in binding energy scores: the tail of the distribution on the positive side reaches nearly a binding score change of 2.0 XEUs and on the negative site values below –0.5 XEUs. Strikingly, the positions in category (**1,1**) yield the most binding enhancing energy scores compared to all other categories, mirroring TEM-1 results. Additionally, the substitutions applied in category (**1,0**) also result in more favorable binding energy scores for dimannose. Mutations in both category (**1,1**) and (**1,0**) present favorable binding energy scores. However, the number of mutations predicted to be enhancing binding in category (**1,1**) is more than those in category (**1,0**) (26% of category (**1,1**) compared to 14% of category (**1,0**)). Interestingly, the mutations in category (**1,0**) that disrupt the binding energy scores is not as strong

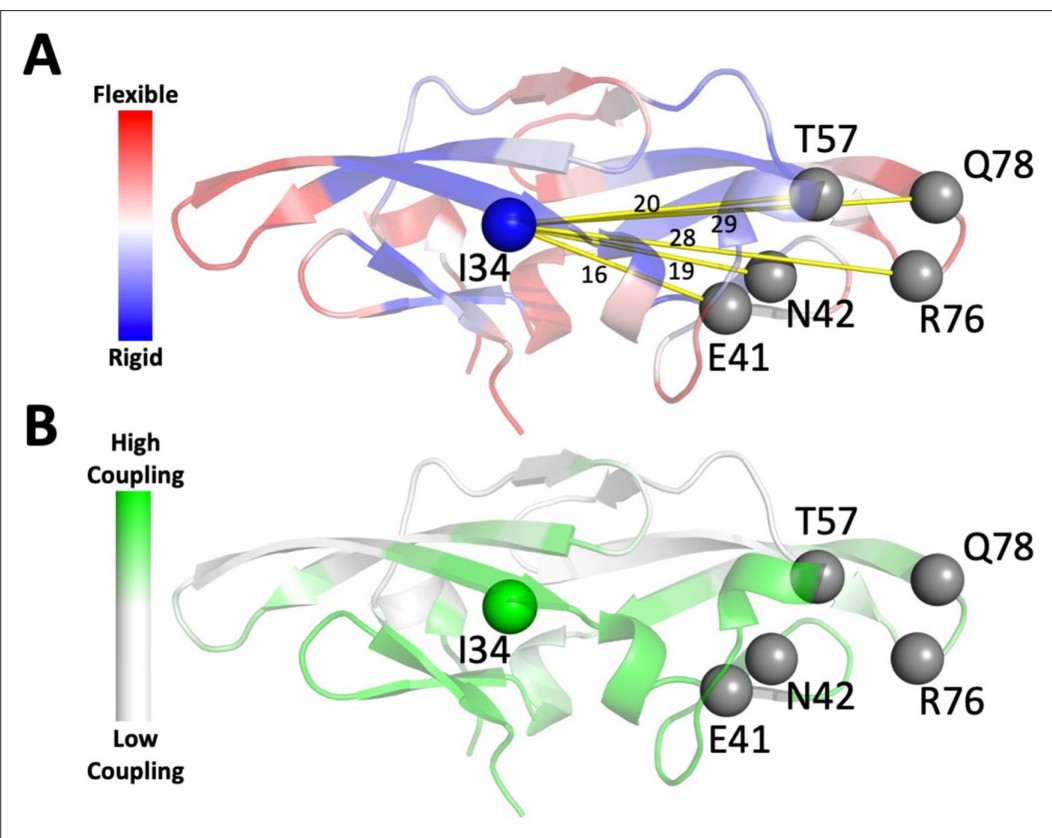

**Figure 3.** DFI and DCI analyses on CV-N. (**A**) Dynamic flexibility index (DFI) profile mapped onto cyanovirin-N (CV-N) structure: red corresponds to high DFI (very flexibile sites), and blue to low DFI values (rigid sites). Position I34 (low DFI score) is highlighted. (**B**) Dynamic coupling index (DCI) profile projected on CV-N structure with green corresponding to sites exhibiting high coupling with binding site residues.

**Table 1.** Predicted binding affinities of domain B, experimental ITC data, and chemical denaturation experiments for P51G-m4 and its I34 variants.

| Protein | Predicted binding score (X-score energy unit) | ITC dimannose $K_d$ (µM) | ITC dimannose $\Delta H$ (kcal/mol) | ITC dimannose $T\Delta S$ (kcal/mol) (T=298 K) | ITC dimannose $\Delta G$ (kcal/mol) | $\Delta G_{H2O}$ (kcal/mol) | $C_m$ (M) |
|---|---|---|---|---|---|---|---|
| P51G-m4 | –6.62 | 117±3 | –12.3±0.3 | –7.00±0.3 | –5.30±0.3 | 3.01±0.047 | 1.46±0.019 |
| P51G-m4-I34K | –5.85 | No binding | No binding | No binding | No binding | 2.40±0.124 | 0.68±0.015 |
| P51G-m4-I34L | –6.19 | 148±2 | –9.60±0.1 | –4.40±0.1 | –5.20±0.1 | 2.95±0.077 | 1.39±0.009 |
| P51G-m4-I34Y | –6.75 | 64±5 | –4.35±0.1 | 1.32±0.2 | –5.67±0.2 | 2.91±0.157 | 1.13±0.017 |

as category (**1,1**), but similar to category (**0,1**) and (**0,0**). The observed mostly neutral behavior with category (**0,0**) agrees with the same trend obtained with TEM-1 analyses.

Overall, the distribution of computational binding scores of dimannose binding to CV-N in each category aligns with the distribution of experimentally characterized TEM-1 fitness results of the same category. However, there are some discrepancies, for example, there are beneficial mutations in category (**0,1**) in TEM-1, but we don't observe the same trend in CV-N. This is due to the initial challenge faced in constructing the MSA of CV-N homologous proteins. There is limited sequence information, and most of the proteins in the CV-N family exhibits binding specificity to a different glycan (*Fujimoto and Green, 2012*; *Koharudin et al., 2009*). In contrast, β-lactamase family proteins exhibit highest activity toward penicillin, and they have been subjected to strong natural selection leading to conservation in both fold and function (*Salverda et al., 2010*; *Zou et al., 2021*). Hence, the less noise in evolutionary analysis in case of β-lactamase family of proteins allows us to correctly filter deleterious type of substitutions based on the MSA. Regardless, however, in both cases, as hypothesized, substitutions on category (**1,1**) residues impact the function most.

To further investigate the mechanism of functional modulation of category (**1,1**) mutations, we chose the position with highest binding enhancing docking scores, I34, from category (**1,1**). I34 exhibits %DFI values lower than 0.2 (*Figure 3A*), is at least 16 Å away from binding residues (distal), dynamically coupled (*Figure 3B*) and co-evolved with the binding pocket (*Supplementary file 2* and *Supplementary file 6*). Moreover, docking scores of I34 variants suggest that the mutations can modulate binding in a wide range: I34Y variant leads to an increase in binding affinity (beneficial), I34K decreases the binding affinity (deleterious), and I34L yields no change (neutral) (*Table 1*).

To verify the predictions of I34 variants, we first assessed the folding and thermal stability of these mutants by circular dichroism (CD) spectroscopy. Far-UV CD spectroscopy showed that all mutants are well folded and adopt a fold similar to the parent protein, characterized by spectra with a single negative band centered at 216 nm. We determined the stability of the mutants by CD monitored thermal denaturation; the thermal denaturation curves were analyzed to obtain apparent melting temperature ($T_m$) values. We found that the conservative mutation I34L is as stable as P51G-m4, with apparent $T_m$ of 57.8°C and 58°C, respectively. In contrast, I34Y and I34K were less thermostable than P51G-m4 as shown by apparent $T_m$ values of 54.7°C and 47°C, respectively. Not surprisingly, substituting a hydrophobic residue with a basic aliphatic amino acid (lysine) has a large destabilizing effect, while aromatic and polar tyrosine is better tolerated. The trend of thermostability is P51G-m4~I34 L> I34 Y> I34 K (*Figure 4—figure supplement 2*).

Chemical denaturation experiments were used to extract thermodynamic values, after ensuring complete equilibration at each concentration of guanidinium hydrochloride by incubating the samples for 72 hr (*Patsalo et al., 2011*). The $\Delta G_{H2O}$ values and $C_m$ values of P51G-m4, I34L, I34Y, and I34K are found as 3.0, 2.94, 2.91, and 2.38 kcal/mol and of 1.45, 1.39, 1.13, and 0.68 M respectively (*Table 1*). The results align with the thermal denaturation results: P51G-m4 is the most stable to denaturant, followed by I34L, I34Y, and I34K (*Figure 4—figure supplement 3*).

Next, we evaluated the impact of the mutations on the dimannose binding affinity by isothermal titration calorimetry (ITC) (*Figure 4—figure supplement 4*); data were analyzed to extract $K_d$ values listed in *Table 1*. We found that I34Y binds dimannose with tightest affinity ($K_d$: 64 µM) of all the mutants tested, a twofold improvement over P51G-m4 ($K_d$: 117 µM). Binding by I34L is slightly weaker

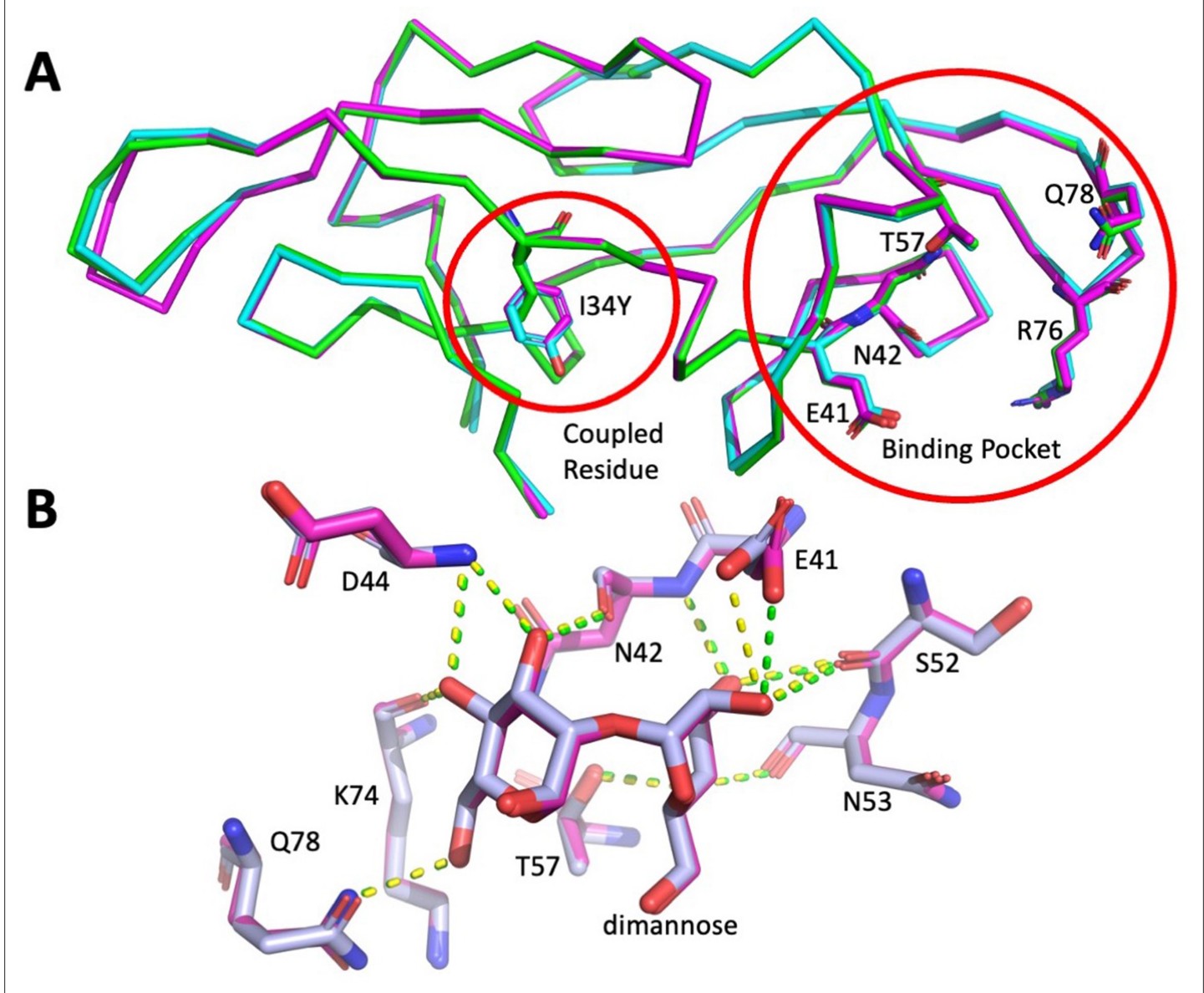

**Figure 4.** The comparison of the crystal structures of P51G-m4 and I34Y. (**A**) The crystal structures of I34Y (bound in magenta and unbound in cyan) and its template protein P51G-m4 (green) are superimposed. (**B**) Overlay of bound structures of I34Y (magenta) and P51G-m4 (gray) (RMSD 0.15 Å); dashed lines depict polar interactions with dimannose.

The online version of this article includes the following figure supplement(s) for figure 4:

**Figure supplement 1.** The flowchart of the Adaptive BP-Dock Scheme.

**Figure supplement 2.** Fits for thermal melts of the cyanovirin-N (CV-N) (**A**) I34 variants and (**B**) A71 variants.

**Figure supplement 3.** Fits for the chemical denaturation experiments of the variants.

**Figure supplement 4.** Binding isotherms of cyanovirin-N (CV-N) mutants upon titration with dimannose: (**A**) I34Y, (**B**) P51G-m4, and (**C**) A71T.

**Figure supplement 5.** Comparison of experimentally solved I34Y structure with docked pose from Adaptive BP-Dock algorithm.

with a $K_d$ of 148 µM. No binding was observed for I34K in these conditions. Thermodynamic values extracted from ITC experiments (*Table 1*), suggesting that entropy changes play an important role in the observed changes in binding affinity: surprisingly, entropy is positive for I34Y, indicating an increase in disorder upon binding.

To glean more information on the mode of binding by I34Y, we determined the X-ray structure of the unbound and dimannose-bound form and compared it with the template protein P51G-m4. The

fold is highly conserved (*Figure 4*) as shown by main chain RMSD of 0.16 and 0.20 Å with bound and unbound I34Y, respectively, and tyrosine is well tolerated at position I34. The binding pocket region is also structurally conserved compared to P51G-m4. Analysis of the polar contacts between dimannose and P51G-m4 and I34Y (*Figure 4B*) shows an identical number of hydrogen bonds (11) with the ligand, indicating a conserved binding pose. We compared the docked pose of I34Y acquired from Adaptive BP-Dock with the bound X-ray structure. The ligand shows an RMSD value of 0.75 Å (*Figure 4—figure supplement 5*). These observations suggest that the increase in binding affinity of I34Y toward dimannose might be mediated by equilibrium dynamics, which are not captured by the crystal structure. This hypothesis is supported by the changes in entropy compensation measured experimentally (ITC) in dimannose binding by P51G-m4 (negative T$\Delta$S) and I34Y (positive T$\Delta$S).

## Molecular mechanism governing the binding dynamics in I34 variants

It is interesting to observe that a distal site can modulate binding affinity to a wide range based on amino acid substitutions. This finding has also been observed for allosterically regulated enzymes such as LacI, for which different amino acid substitutions on non-conserved sites lead to gradual changes in function, acting like a rheostatic switch to modulate function through conformational dynamics (*Campitelli et al., 2021*; *Campitelli et al., 2020b*; *Meinhardt et al., 2013*; *Miller et al., 2017*; *Swint-Kruse et al., 1998*). To gather atomic level detail on how the substitutions on I34 dynamically modulate the binding affinity, we employed MD simulations in both bound and unbound forms (see Methods for details of the simulations). The unbound trajectories were analyzed for acquiring binding pocket hydrogen bond distances and pocket volume. Later, to learn about the ligand-induced conformational dynamic changes, the bound trajectories were utilized to estimate computational binding free energies (*Deng and Roux, 2009*; *Okazaki et al., 2006*).

Previous computational work in our lab had linked binding affinity in the CV-N family to the accessibility of the binding pocket: A hydrogen bond between the amide hydrogen of N42 and carbonyl oxygen of N53 forms a closed pocket, hindering glycan accessibility, whereas the loss of this hydrogen bond leads to an open pocket (*Li et al., 2015*). Using the formation of this hydrogen bond in the trajectories of unbound WT and I34Y as metric for assessing open and closed conformations, we found that I34Y variant samples the open binding pocket more often than P51G-m4 (*Figure 5—figure supplement 1*).

Another compelling evidence differentiating I34 variants from P51G-m4 is the change in their binding pocket volumes estimated by POVME pocket volume calculation tool (*Wagner et al., 2017*). The calculated pocket volumes for I34Y, I34K, and P51G-m4 were converted into frequencies to obtain probability distributions (*Figure 5A*), revealing that I34Y variant samples a more compact pocket volume compared to P51G-m4. If the pocket is too small or too large, dimannose cannot maximize its interaction with the protein, and a compact conformation enables dimannose to easily make the necessary hydrogen bond interactions with the protein. This optimum pocket volume sampled by I34Y may also explain the different binding energetics observed by ITC, in which a positive entropy change upon binding compensates for the loss in enthalpy compared to P51G-m4 (*Table 1*; *Breiten et al., 2013*; *Cornish-Bowden, 2002*). Pocket volume analysis reveals a larger value for I34K compared to P51G-m4, suggesting that this mutant cannot accommodate the necessary interactions with the dimannose resulting in loss of binding. We applied the same pocket volume calculation to the X-ray structures of P51G-m4 and I34Y variant, and we found volumes of 141 and 114 Å$^3$ for P51G-m4 and I34Y, respectively, in the unbound forms (*Figure 5B*). These volumes correlate well with the mean volumes from MD trajectories, suggesting that the variants modulate the conformational dynamics of binding pocket.

Overall, the conformational dynamics analysis of the unbound conformations indicates a shift of the native ensemble toward a smaller pocket volume upon I34Y mutation. This could explain the decrease in the entropic cost of binding observed in ITC results. We also analyzed the binding energetics by carrying out dimannose docking with 2000 different conformations sampled from the binding pocket volume distributions. We found that the small volume restrict accessibility to the side-chain conformations of binding residue R76 in the I34Y variant, yielding different hydrogen bond patterns with the dimannose (*Figure 5—figure supplement 2*) and suggesting a loss in enthalpic contribution.

The bound simulation trajectories were subjected to the MM-PBSA approach to estimate computational binding free energies and related enthalpic and entropic contributions (*He et al., 2020*;

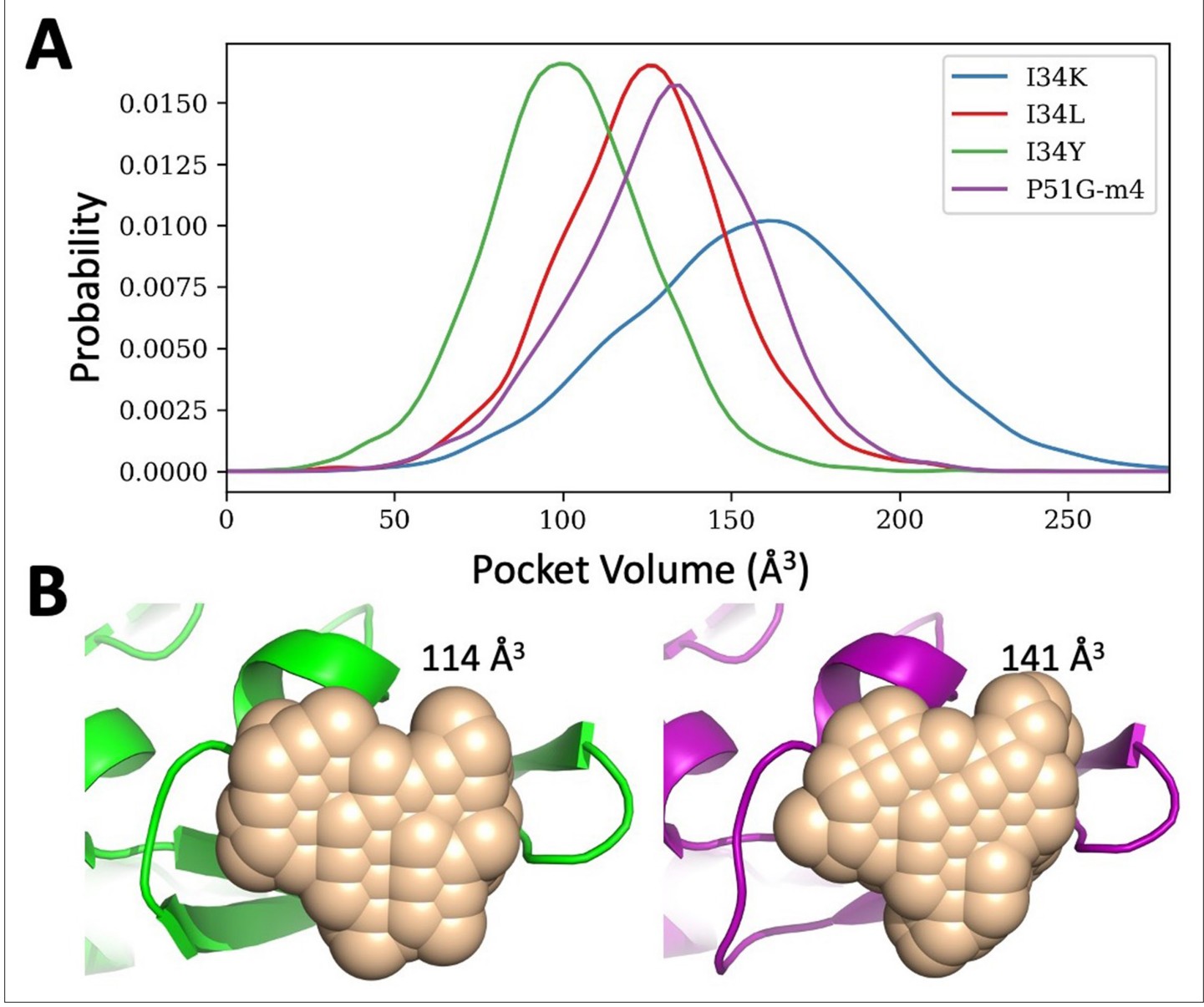

**Figure 5.** Binding pocket volume estimations for P51G-m4 and its variants. (**A**) Probability distribution of the pocket volume analyses obtained from molecular dynamics (MD) simulation trajectories. I34Y populates a conformation with an optimum volume more than others. P51G-m4 and I34L variant samples similar pocket volumes, but I34K variant has a larger pocket volume compared to others. (**B**) Pocket volume comparison of the domain B of solved structures for P51G-m4 (purple) and I34Y variant (green).

The online version of this article includes the following figure supplement(s) for figure 5:

**Figure supplement 1.** The difference in accessibility of the binding pocket for P51G-m4 and I34Y.

**Figure supplement 2.** We sampled 2000 different conformations from molecular dynamics (MD) simulations for P51G-m4 cyanovirin-N (CV-N) and I34Y mutant and performed dimannose docking to obtained docked poses and then analyzed hydrogen bond patterns.

*Rastelli et al., 2010*). The results are tabulated on *Supplementary file 4*. The computed binding free energies capture the trend of experimental binding affinities (R=0.87). The I34Y variant displays a more favorable binding with dimannose compared to wild type. Interestingly, both experimental and computational results show I34Y compensating the enthalpic loss with entropic gain. While I34L variant enthalpic loss is greater than I34Y in computational approach, the overall binding free energy mirrors the ITC results. Additionally, loss of binding of I34K variant overlaps with the ITC data.

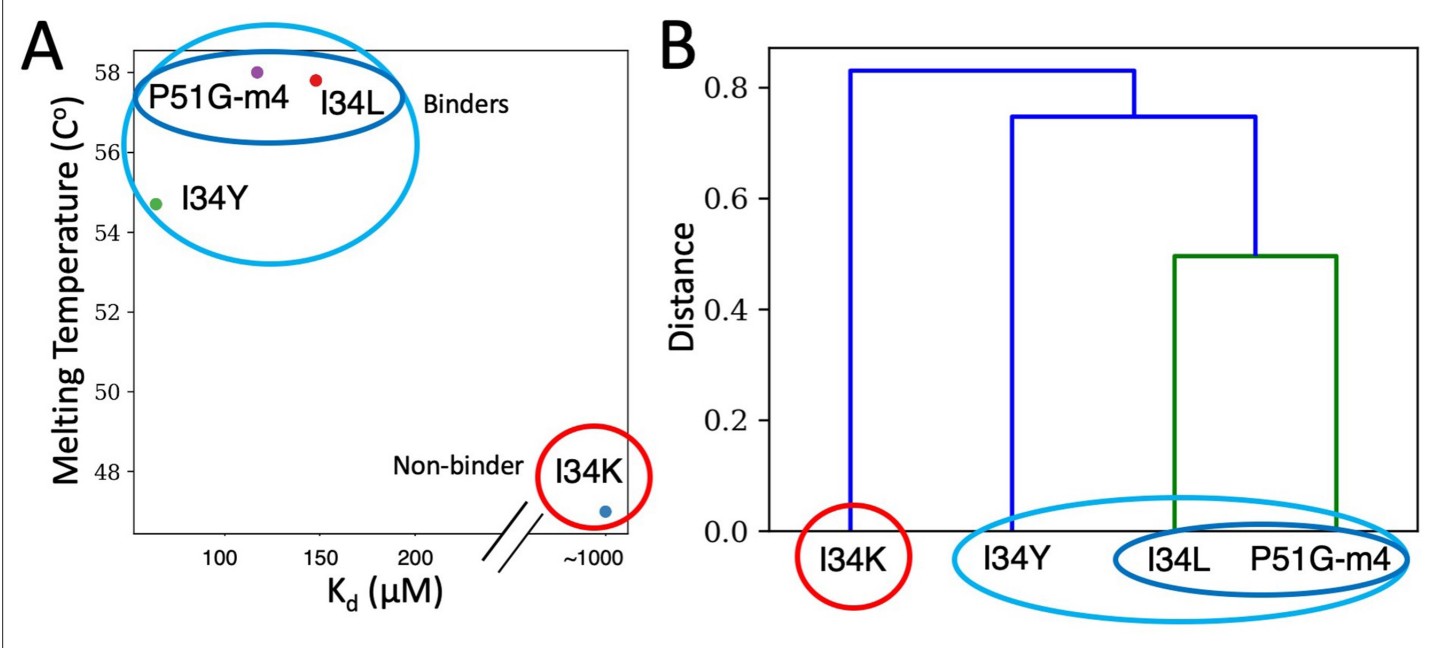

**Figure 6.** Clustering of CV-N variants using DFI profiles and biophysical properties.
(**A**) 2D map of $K_d$ and melting temperature of P51G-m4 and its variants. (**B**) Principal component analyses (PCA) clustering on the first two principal components of the dynamic flexibility index (DFI) profiles as a dendrogram.

## Substitutions of I34 modulates the conformational ensemble leading to change in dimannose binding affinity

Proteins adapt to a new environment by modulating the native state ensemble through mutations of different positions while keeping the 3D structure conserved (*Campitelli et al., 2020a*; *Kuriyan and Eisenberg, 2007*; *Li et al., 2015*; *Liu and Nussinov, 2017*; *Modi and Ozkan, 2018*; *Risso et al., 2018*; *Tripathi et al., 2015*; *Woodrum et al., 2013*). As we also observed a similar pattern of conservation of structure yet change in function in our designed CV-N I34 variants, we further analyzed the flexibility profiles of I34 variants. The DFI profiles clustered using principal component analyses match the 2D map of melting temperature and $K_d$ as reaction coordinates, suggesting a correlation between changes in dynamics and changes in function (*Figure 6*). The 2D map shows I34L, P51G-m4, and I34Y under the same cluster, with I34L and P51G-m4 close, while I34K is markedly different (*Figure 6A*). The dendrogram constructed based on the DFI profiles captures this clustering (*Figure 6B*) with P51G-m4 and I34L variant under the same branch, suggesting their dynamics are very similar; I34Y is under the same main cluster albeit in a different branch. I34K is under a separate branch, indicating different dynamics. This is in agreement with our previous studies, where substitutions on DARC spots modulate binding dynamics reflected in their flexibility profiles to adapt to a new environment (*Campitelli et al., 2021*; *Kumar et al., 2015b*; *Modi et al., 2021a*).

We further gleaned a molecular view of the role of flexibility in binding by comparing changes in DFI profiles of the binding site residues with P51G-m4 for each mutant, in the unbound and bound form (*Figure 7A and B*). We found that flexibility at position T57 is highly dependent on the amino acid at position I34: flexibility increases in I34K, suggesting a higher entropic penalty for binding interactions. It is unchanged in I34L, which has similar binding affinity. In contrast, T57 becomes much more rigid in I34Y mutant. This indicates that the rigidification leading to a decrease in the entropic cost can contribute to the binding affinity enhancement of this mutant, which is also in agreement with the ITC results.

Comparison of the flexibility profiles of the bound form with those of the unbound form reveals that residue I34 in WT drastically gets rigidified upon binding, whereas I34Y variant does not. The decreased flexibility of T57 in the unbound form of I34Y accommodates the interactions with dimannose, contributing to the entropic compensation. In addition to the binding site residues of domain B, the flexibility of the rest of the residues also contributes to the total change in binding free energies.

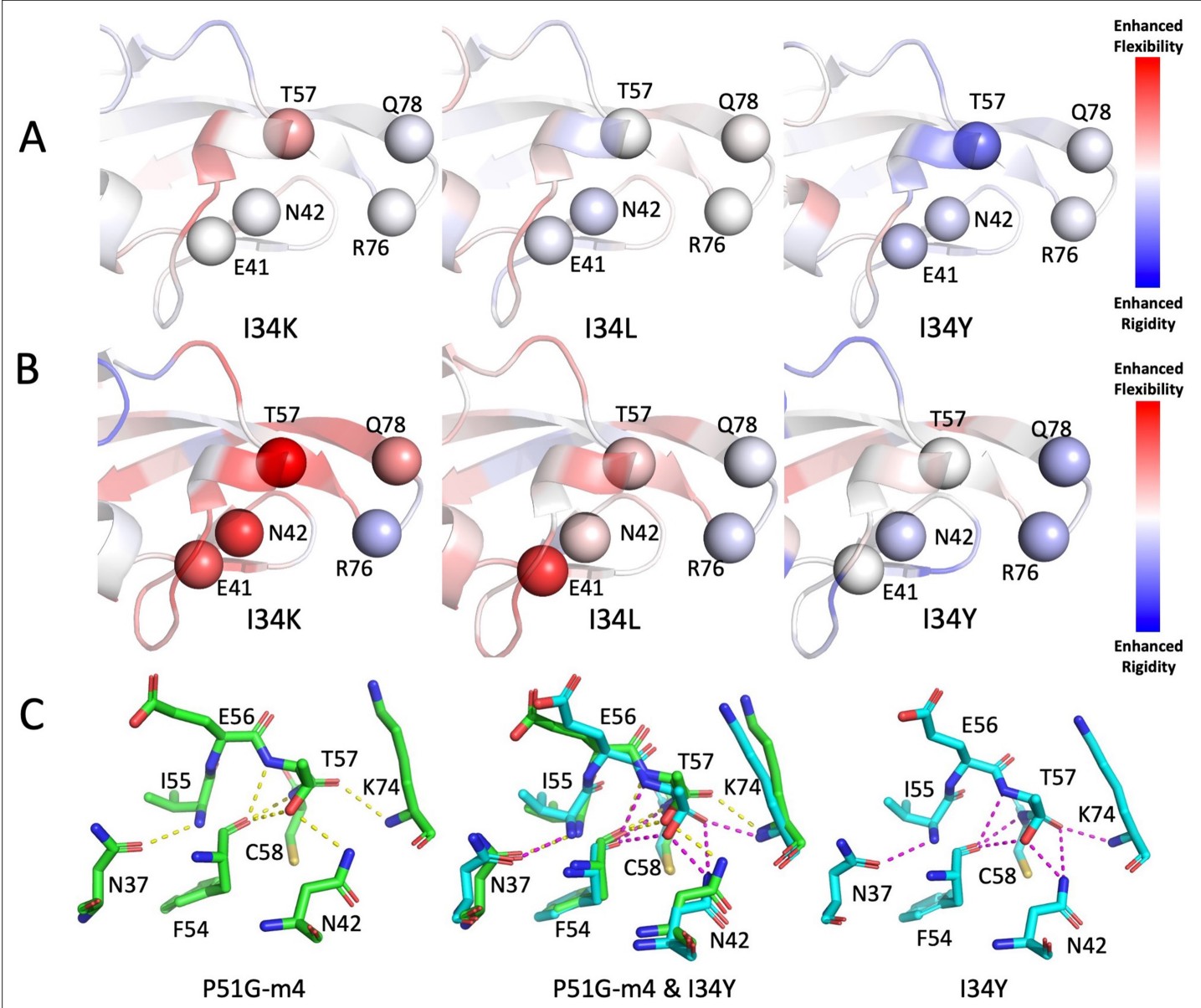

**Figure 7.** Changes inflexibility s of the binding site residues upon mutations in bound and unbound forms. (**A**) Change in flexibility of I34K, I34L, and I34Y relative to P51G-m4 in unbound form is shown. Residues E41, N42, and T57 rigidifies on I34Y compared to P51G-m4. (**B**) Change in flexibility of I34K, I34L, and I34Y relative to P51G-m4 in bound form is projected on structure. (**C**) Hydrogen bonding interactions of residues I55, E56, T57, and C58 are shown for P51G-m4 and I34Y variant.

The online version of this article includes the following figure supplement(s) for figure 7:

**Figure supplement 1.** Correlation between change in DFI profiles aind change in ΔG of binding.

**Figure supplement 2.** Network of hydrogen bond interactions connecting residue location 34 to T57 is investigated in I34Y variant and P51G-m4 cyanovirin-N (CV-N).

Therefore, we analyzed the correlation between (i) the sum of total change in flexibility of the binding site residues, (ii) the binding site residues and the residues exhibiting highly coupling with the binding pocket, with the experimentally measured binding affinity change. We observe a strong correlation between change in flexibility and change in affinity, as expected I34Y exhibiting tighter binding also gets more rigidified upon binding compared to P51G-m4. Moreover, inclusion of the highly coupled residues in addition to the domain B binding sites in computing the total sum of DFI scores yields a higher correlation with the experimental binding affinity change (*Figure 7—figure supplement 1A*).

On the other hand, the correlation between the flexibility change of the randomly selected residues and experimental binding affinities yields poor correlation coefficient (*Figure 7—figure supplement 1B*). These results strongly support the role of dynamic allostery in modulating binding affinity.

The rigidification of T57 in I34Y variant is compelling evidence that the distal mutation is allosterically controlling the binding site dynamics. We further computed the network of interactions that connects the residue position 34–57 and investigated whether distinct pathways emerge after I34Y mutation. We analyzed the hydrogen bond networks, particularly computed the possible network of hydrogen bonds creating pathways from 34 to 57 using the sampled snapshots from the MD trajectories (*Figure 7—figure supplement 2*). This analysis presents a unique pathway from 34 to 57 by first forming a new hydrogen bond between the side chain oxygen of the Tyrosine 34 and the nitrogen of the Tyrosine 100 in I34Y variant. Furthermore, a second pathway is also found which is sampled much more frequently in I34Y variant strengthening the communication between positions 34 and 57. Thus, both pathways may contribute to the rigidification of T57. We also analyzed the conformations from MD clustered with highest percentage based on alpha carbon RMSD for I34Y and P51G-m4 and compared the hydrogen bond interactions of T57 and its neighboring residues. The closest neighbors of T57; positions I55, E56, and C58 conserved their hydrogen bond interactions with their surrounding residues between P51G-m4 and I34Y. On the other hand, T57 makes an additional hydrogen bond interaction in I34Y compared to P51G-m4 (*Figure 7C*), suggesting that enhancement in hydrogen bond networking of T57 in I34Y leads to rigidification of this position in equilibrium dynamics.

To gain more insight on distal dynamic modulation of binding pocket particularly the decrease of binding site flexibility through distal coupling, we computationally and experimentally characterized another residue, A71, belonging to category (**1,1**) and its mutations: T, S. The docking scores and DFI profiles of A71 variants show high similarity to position I34 ones. The variant A71T is predicted as binding enhancing by our docking scheme displaying a similar binding score as I34Y (A71T predicted binding score: –6.81 XEU), whereas variant A71S is predicted as analogous to I34L variant (A71S predicted binding score: –6.20 XEU). This position is next to residue E72, which is within the hydrogen bond pathway (Pathway 2) (*Figure 7—figure supplement 2*) identified previously connecting I34 and binding residue T57. Furthermore, the computed binding free energies by MM-PBSA is found to be correlating with position I34 results. The A71T variant shows a binding free energy near I34Y (A71T $\Delta$G: –13.70 kcal/mol with $\Delta$H: –29.33 kcal/mol and T$\Delta$S: –15.63 kcal/mol), and A71S close to I34L (A71S $\Delta$G: –9.98 kcal/mol with $\Delta$H: –29.00 kcal/mol and T$\Delta$S: –19.02 kcal/mol). All computational analyses suggested that A71 can modulate binding affinity through distal dynamic coupling similar to I34, hence we experimentally characterized these two variants.

The experimental binding affinity by ITC correlates with in silico predictions. When the change in total DFI score upon binding is compared to change in free energy of binding from ITC experiments (*Figure 7—figure supplement 1*), A71T ($\Delta$G: –5.70 kcal/mol with $\Delta$H: –6.00 kcal/mol and T$\Delta$S: –0.30) features both a change in total DFI and $\Delta$G closer to I34Y, and A71S (A71S $\Delta$G: –5.10 kcal/mol with $\Delta$H: –9.10 kcal/mol and T$\Delta$S: –4.00) shows a score identical to I34L. The entropy of A71T shows a similar change as I34Y experimentally (A71T T$\Delta$S: –0.30) indicating that the same compensation mechanism is utilized by another category (**1,1**) residue. A71S is closer to I34L (A71S T$\Delta$S: –4.00). Similar to I34Y, the melting temperature of A71T is lower than P51G-m4 (*Figure 4—figure supplement 2*). Results of A71 variants further establish the potential of ICDC and category (**1,1**) residues in diversely tuning the binding affinity of domain B of CV-N through playing enthalpy-entropy compensation of binding process.

Our new ICDC approach suggests that it is possible to identify and incorporate distal mutations into protein design bringing together evolutionary inferences with long-range dynamic communications within the 3D network of interactions.

## Methods
### Adaptive BP-Dock

Adaptive backbone perturbation docking, Adaptive BP-Dock in short, allows us to model the interaction between CV-N and glycans in silico (*Bolia and Ozkan, 2016*). Adaptive BP-Dock combines the complex simulation of backbone flexibility of a protein into Rosetta's ligand docking application (*Davis and Baker, 2009*). The common restriction in docking is the implementation of flexibility

of receptor and ligand (*Davis et al., 2009*; *Davis and Baker, 2009*; *DeLuca et al., 2015*; *Meiler and Baker, 2006*). Rosetta included the flexibility of ligand in their Monte Carlo sampling approach but lacking full receptor flexibility. This high-order challenge is overcome by utilizing perturbation response scanning (PRS) to compute backbone changes during docking (*Atilgan and Atilgan, 2009*; *Bolia et al., 2014b*; *Ikeguchi et al., 2005*). This procedure also allows the modeling of transition from an unbound state to a bound state (*Bolia and Ozkan, 2016*). The computational cost of sampling is reduced by using a coarse-grained approach employing elastic network model (ENM) leading to an efficient way of computing backbone perturbations, mimicking the ligand interacting with receptor (*Atilgan et al., 2001*; *Atilgan et al., 2010*; *Atilgan and Atilgan, 2009*).

We employed Adaptive BP-Dock in modeling glycan CV-N interactions starting from an unbound conformation of CV-N. The perturbed pose of the protein is calculated using PRS. The structure is then minimized, and the side chains are added at this step. The glycan is docked to the minimized structure using RosettaLigand algorithm. Rosetta samples bound conformations using a knowledge-based potential function and calculates bound pose energies. The lowest energy docked pose is selected and feed back to perturbation step, and the same procedure is followed iteratively until a convergence is reached. At the end of each iteration the lowest energy docked pose is taken and binding score is calculated using an empirical scoring function X-score. XEUs have shown to provide higher correlations with experimental results (*Wang et al., 2002*). The flow of the algorithm is shown in *Figure 4—figure supplement 1*. Adaptive BP-Docks iterative algorithm ensures that the sampling does not get trapped in a local minimum and reaches a global minimum. The challenge of unbound/bound modeling is solved using the iterative approach as the conformations are led toward a bound pose with the help of PRS.

## Molecular dynamics

Gromacs simulations are conducted for P51G-m4 CV-N and all the variants in unbound form, and further for P51G-m4 CV-N, I34 variants I34K, I34L, I34Y, and A71 variants A71S, A7T in bound form (*Abraham et al., 2015*; *Van Der Spoel et al., 2005*). For each simulation the all-atom system is parametrized with CHARMM36 force field and explicit water model TIP3P. The solvation box is set to be minimum 16 Å from the edge of the protein. The system is neutralized by potassium ions to sustain electroneutrality and minimized with steepest descent for 10,000 steps. A short-restrained equilibrium is conducted in the constant number of particles, pressure, and temperature ensemble (NPT) for 5 ns using the Berendsen method at 300 K temperature and 1 bar pressure. NPT production trajectories were performed with Nose-Hoover and Parrinello-Rahman temperature and pressure coupling methods for 2 μs at 300 K and 1 bar. For all cases periodic boundary conditions and particle-mesh Ewald (PME) with interaction cutoff of 12 Å is employed with Gromacs version 2018.1.

## Dynamic flexibility index

DFI is a position-specific metric that can measure the resilience of a given position to the force perturbations in a protein. It calculates the fluctuation response of a residue relative to the gross fluctuation response of the protein (*Kumar et al., 2015b*; *Larrimore et al., 2017*). DFI calculates residue response due to a perturbation by utilizing covariance matrices.

$$[\Delta R]_{3Nx1} = [H]_{3Nx3N}^{-1}[F]_{3Nx1}$$

$$DFI_i = \frac{\sum_{j=1}^{N} |\Delta R^j|_i}{\sum_{i=1}^{N}\sum_{j=1}^{N} |\Delta R^j|_i}$$

Residue response, Δ**R**, is calculated using linear response theory by applying force, **F**, in multiple directions to mimic isotropic fluctuations. Hessian matrix, **H**, contains second derivatives of potentials. The inverse of Hessian matrix, **H⁻¹**, contains residue covariances, and interpreted as a covariance matrix. The covariance matrices can be gathered from MD simulations, and also by using ENM of a protein. In this study, MD covariance matrices have been utilized to incorporate residue interactions accurately.

Residues with low DFI score (below 0.2) are considered as hinge points. These points are communication hubs in this 3D interaction network. Due to high coordination number, the residues exhibiting low DFI values are crucial as information gateways. While they do not exhibit high residue fluctuation to the perturbations, they quickly transfer the perturbation information to other parts, thus they are in control of collective motion of the protein. A change in low DFI positions (i.e., a mutation) will lead to a transformation in the communication grid and majority of disease-associated (i.e., function altering mutations) are often observed as hinges (*Butler et al., 2015*; *Nevin Gerek et al., 2013*; *Kumar et al., 2015a*). The substitution on these site usually alters catalytic activity or binding interaction (i.e., glycans) by modulating equilibrium dynamics (*Campitelli et al., 2020a*).

## Dynamic coupling index

DCI exploits the same framework of DFI (*Campitelli et al., 2020a*; *Larrimore et al., 2017*). DCI utilizes the residue response fluctuation upon random force perturbation at a specific residue position to investigate residues that exhibit long-range coupling to each other. In DCI approach, a unit force is applied on functional residues (i.e., binding site residues) one by one and responses of all other residues are calculated.

$$DCI_i = \frac{\sum_{J}^{N_{Functional}} |\Delta R^j|_i / N_{Functional}}{\sum_{J=1}^{N} |\Delta R^j|_i / N}$$

With DCI scheme the residues with high response (high DCI score) indicate high long-range dynamic coupling. Residues with high DCI values with binding sites play a critical role in intercommunication of a protein with the binding residues. These coupled residues are of utmost importance in how forces propagate through amino acid chain network on a binding event. Some of the coupled residues are far from the binding site but still encompass modulation capabilities over binding pocket.

## Informing dynamics from co-evolution

Co-evolutionary data paves the way to assessing 3D structural contacts by utilizing available sequence information (*Hopf et al., 2018*; *Marks et al., 2012*; *Morcos et al., 2014*). Sequence information is more abundant compared to resolved protein structures. Exploiting the sequence information, primary contacts comparable to realistic structural contacts can be calculated and a contact matrix is formed. The accuracy of these contact maps is proved to be valuable in protein folding studies (*Kryshtafovych et al., 2019*; *Morcos et al., 2011*; *Wang et al., 2016*). EC analysis is used to collect information on how much two residues in a protein sequence are in close proximity in 3D structure. EC scores could be calculated by many different statistical approaches. In this study EC information is gathered by using RaptorX, EVcouplings, and MISTIC webservers (*Hopf et al., 2019*; *Simonetti et al., 2013*; *Wang et al., 2017*). While the limitation of these methods emerges from sequence homolog availability of a protein in MSA, RaptorX uses a deep neural network leveraging joint family approach, combining multiple ortholog protein families sharing similar function and phylogeny, to infer possible contacts. This method is proven to produce high accuracy in contact prediction compared to others (*Wang et al., 2017*). However, for a given MSA containing enough homolog sequences, other methods are also strong in predicting spatial contacts. EVcouplings approach uses direct information (DI) to calculate co-EC. DI metric is a modified mutual information (MI) score considering consistency between pairwise probabilities and single amino acid frequencies (*de Juan et al., 2013*; *Morcos et al., 2011*). Nonetheless, MI, a global approach compared to local DI metric, is accurate in capturing true contacts, while entangling indirect contacts from direct contacts. MISTIC webserver has taken advantage of MI to calculate co-EC (*Dunn et al., 2008*; *Gouveia-Oliveira and Pedersen, 2007*; *Simonetti et al., 2013*). In their MI method they introduced a correction term to MI to surpass the low statistics gathered with an MSA containing limited number of sequences. This approach is very useful in cases where certain homologs are rare and MSA of these homologs have multiple gaps in their alignments. All of these methods are employed in this study to achieve high accuracy predictions in finding residue couplings.

## Mutant proteins cloning, expression, and purification

The genes for mutants (I34Y, I34K, and I34L) were generated by applying mutagenic primers to P51G-m4-gene sequence and amplifying by PCR. The constructs were subsequently cloned in pET26B vector between NdeI and XhoI sites and transformed in BL21(DE3) for expression and purification. The proteins were expressed from a 10 ml starter culture in LB broth overnight at 37°C, inoculated into 1 l LB medium. The culture was induced with 1 mM isopropyl thiogalactoside when OD reached 0.6 and grown for another 6–8 hr. Then, the cells were harvested by centrifugation, lysed in 6 M guanidine hydrochloride at pH 8.0, and sonicated for 10 min. The supernatant recovered after centrifugation was used to purify proteins with GE HisTrap HP column (GE Healthcare Bio-Sciences, Piscataway, NJ) and a Bio-Rad EconoPump (Bio-Rad, Richmond, CA) under denaturing conditions. In brief, the proteins were loaded on the column in Gu-HCl buffer, which was buffer exchanged by 8 M urea buffer. The nonspecific proteins were washed out by 4 M urea and 20 mM imidazole buffer, pH 8.0 and eluted with 2 M urea and 200 mM imidazole, pH 8.0 buffer before putting it for overnight dialysis against 10 mM Tris pH 8.0 and 100 mM NaCl buffer. The buffer was changed once during the night. The refolded protein was concentrated and re-purified to isolate the monomeric species by size exclusion chromatography using Sephadex 75 10/300 column on Agilent's Infinity 1260 system. The gel filtered protein was finally used for all the experiments.

## CD spectroscopy and T-melts

In CV-N family proteins, thermodynamic parameters like free energy of unfolding, enthalpy, and entropy cannot be extracted by thermal denaturation because the transition from folded to unfolded state is non-reversible (*Patsalo et al., 2011*), therefore melting temperatures are used. Far-UV CD spectra were recorded on a Jasco J-815 spectropolarimeter equipped with a thermostatic cell holder, PTC 424S. Spectra were measured from 250 to 200 nm, using a scanning speed of 50 nm/min and a data pitch of 1.0 nm at 25°C. Samples concentration was approximately 15 µM in 10 mM Tris, pH 8.0, and 100 mM NaCl. For thermal denaturation experiments, the melting profile was monitored at 202 nm from 25°C to 90°C. The data points were plotted and fitted in Origin8.5 software to get apparent Tm.

## Isothermal titration calorimetry

ITC was performed at the Sanford-Burnham Medical Research Institute Protein Analysis Facility using ITC200 calorimeter from Microcal (Northampton, MA) at 23°C; 2.0 µl aliquots of solution containing between 3 and 10 mM Man2 were injected into the cell containing between 0.057 and 0.11 mM protein. Nineteen of 2.0 µl injections were made. The experiments were performed in 10 mM Tris, 100 mM NaCl, pH 8.0 buffer. ITC data were analyzed using Origin software provided by Microcal.

## Chemical denaturation experiments

Chemical denaturation experiments were done by monitoring the shift in the intrinsic tryptophan fluorescence on Cary Eclipse instrument (Varian). Ten µM of protein samples were incubated with increasing concentrations of guanidine hydrochloride in the range of 0–6 M in 50 mM Tris pH 8.0 buffer for 72 hr at 25°C. The emission spectra for the same were recorded by keeping the excitation wavelength at 295 nm and bandwidth of 1 nm. A ratio of fluorescence at 330 and 360 nm ($I_{330}/I_{360}$) was plotted at respective Gu-HCl concentrations, and the data points were fit to following sigmoidal equation to obtain $C_m$.

$$y = A2 + \frac{A1 - A2}{1 + e^{(x - x0)/dx}}$$

where A1 and A2 are the initial and final 330/360 ratios and x0 is the concentration of Gu-HCl, where y = (A1+A2)/2, or the point, where 50% of the population is unfolded. It is also denoted as $C_m$.

The denaturation curve was used to calculate the free energy of the protein in the absence of denaturant ($\Delta G_{H2O}$). Fraction unfolded ($f_U$) was calculated using the following formula:

$$f_U = (Y_F - Y_{obs})/(Y_F - Y_U)$$

where $f_U$ is the fraction unfolded, $y_F$ is the value when there is no denaturant, $y_{obs}$ is the value at each position, and $y_U$ is the value for unfolded protein. Since $f_U + f_F = 1$, the equilibrium constant, K, of the free energy change can be calculated using

$K = f_U / f_F$

$K = f_U / 1 - f_F$

$\Delta G = -RT \ln K$

where R is the gas constant whose value is 1.987 cal/mol·K and T is the temperature of incubation, which was 298 K. The value of $\Delta G$ is linear over a limited range of Gu-HCl. The linear fit over that range was extrapolated to obtain $\Delta G_{H2O}$.

## Crystallization and structure determination

I34Y was purified as discussed previously and the monomeric gel filtered protein was concentrated to 8 mg/ml. We got the crystals in 2 M ammonium sulphate and 5% (v/v) 2-propanol after screening it in Index HT screen from Hampton Research. The protein crystals were reproduced using same condition in hanging drop method. For protein crystals with dimannose, the crystals were incubated in 1.2-fold molar excess of dimannose. Single needle-like crystals were picked up and cryo-preserved in 25% glycerol before freezing them for data collection at Synchrotron ALS, beamline 8.2.1. Single crystal diffraction was measured at wavelength of 0.999 A with ADSC quantum 315r detector. The data were evaluated to resolution of 1.25 A. The data acquired was indexed using XDS and scaled by the aimless package from CCP4i program suite. The structural coordinates and phase were determined by molecular replacement using 2RDK PDB code. The structure of I34Y of CV-N is deposited under PDB accession code 6X7H. The structure was further refined in Coot.

## Acknowledgements

SBO acknowledges support from the Gordon and Betty Moore Foundations and National Science Foundation (Award: 1715591 and 1901709). This work was supported in part by NIH award 1R21CA207832-01.

## Additional information

### Funding

| Funder | Grant reference number | Author |
| --- | --- | --- |
| National Science Foundation | 1715591 | S Banu Ozkan |
| Gordon and Betty Moore Foundation | 1901709 | S Banu Ozkan |
| National Institutes of Health | 1R21CA207832-01 | Giovanna Ghirlanda S Banu Ozkan |

The funders had no role in study design, data collection and interpretation, or the decision to submit the work for publication.

### Author contributions

I Can Kazan, Conceptualization, Resources, Data curation, Software, Formal analysis, Validation, Investigation, Visualization, Methodology, Writing – original draft, Writing – review and editing; Prerna Sharma, Conceptualization, Resources, Data curation, Formal analysis, Validation, Investigation, Visualization, Methodology, Writing – original draft, Writing – review and editing; Mohammad Imtiazur Rahman, Resources, Formal analysis, Validation, Investigation, Visualization, Writing – review and editing; Andrey Bobkov, Raimund Fromme, Formal analysis; Giovanna Ghirlanda, S Banu Ozkan, Conceptualization, Resources, Supervision, Funding acquisition, Methodology, Writing – original draft, Project administration, Writing – review and editing

## Author ORCIDs

I Can Kazan http://orcid.org/0000-0003-2593-4179
Giovanna Ghirlanda http://orcid.org/0000-0001-5470-1484
S Banu Ozkan http://orcid.org/0000-0002-9351-3758

## Decision letter and Author response

Decision letter https://doi.org/10.7554/eLife.67474.sa1
Author response https://doi.org/10.7554/eLife.67474.sa2

## Additional files

### Supplementary files

• Transparent reporting form

• Supplementary file 1. Dynamic flexibility index (DFI), dynamic coupling index (DCI), RaptorX, Evcoupling, and MISTIC metrics are used to identify residues in TEM-1 β-lactamase for the four unique categories (*Supplementary file 5*).

• Supplementary file 2. Dynamic flexibility index (DFI), dynamic coupling index (DCI), RaptorX, Evcoupling, and MISTIC metrics are used to identify residues in cyanovirin-N (CV-N) for the integrated co-evolution and dynamic coupling (ICDC) categories (*Supplementary file 6*).

• Supplementary file 3. The predicted binding affinities of domain B and comparison with experimental isothermal titration calorimetry (ITC) data for wild type, mutDB, and P51G-m4 benchmarking.

• Supplementary file 4. Binding free energies, enthalpy and entropy values for wild-type cyanovirin-N (CV-N) and its variants calculated with MM-PBSA approach applied on dimannose bound molecular dynamics (MD) simulations.

• Supplementary file 5. Contains the complete TEM-1 dynamic flexibility index (DFI), dynamic coupling index (DCI), RaptorX, Evcoupling, and MISTIC metric data used in this study.

• Supplementary file 6. Contains the complete cyanovirin-N (CV-N) dynamic flexibility index (DFI), dynamic coupling index (DCI), RaptorX, Evcoupling, and MISTIC metric data used in this study.

### Data availability

All data generated or analysed during this study are included in the manuscript and supporting files.

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
