## [Editor Report]

A computational approach is proposed to identify mutations in enzymes that might impact their interactions with substrates. For one enzyme, in particular, the predictions are validated through experiments, using multiple techniques. Taken together, these data lead to non-trivial conclusions in regard to the nature of allosteric effects, albeit it remains unclear whether these conclusions will apply more broadly when other enzymes are examined.

---

## [Decision Letter]

**Decision letter after peer review:**

Thank you for submitting your article "Design of novel CV-N variants by modulation of binding dynamics through distal mutations" for consideration by *eLife*. Your article has been reviewed by 2 peer reviewers, including Nir Ben-Tal as the Reviewing Editor and Reviewer #1, and the evaluation has been overseen by Michael Marletta as the Senior Editor.

Kazan and colleagues have developed a computational approach to identify distal/remote residues that allosterically modulate dynamics of binding sites. They have utilized a combined bioinformatics (statistical analysis of coevolution) and structural dynamics (looking for dynamic coupling) approach to achieve this. The work that is done to analyze and rationalize the effects of mutations at position 34 f the cyanovirin-N protein are comprehensive and the results are interesting. However, this is not sufficient to test a new technique or approach. There should be a considerably higher number of tests, at different positions and in different proteins, and it should have as a control tests with residues selected through coevolutionary analysis alone, or solely through structural analysis (or picked at random). This would help to understand how or why the approach works (or if it is comparably effective to simply selecting residues at consensus positions). We would also suggest some statistical analysis to estimate how accurate the approach is. The concern is essentially that with n=1, as is the case here, we cannot properly test the hypothesis that the approach as outlined in the paper is effective. We feel it is an interesting paper, but simply needs more data – without it, it is very speculative.

The explanation in terms of entropy is overly simplistic and is focused fully on the protein dynamics. The ITC measures the binding thermodynamics of the systems, which includes solvent. In this case it is likely the entropy reflects release of bound waters – without understanding how the solvation of the protein changes, especially in the binding site upon ligand binding a significant part of the explanation is missing. This could in fact help explain some of the differences in effects of the mutations. This could be looked at through the MD simulations.

In summary, the main strength is that the predictions, obtained by combining various computational tools, guided experiments, again using multitude of techniques. It all converged to interesting conclusion about allosteric effects of mutations on substrate binding which are not trivial. However, there are some unclear and open-ended issues that should be dealt with.

Essential revisions:

Summary

A computational approach is used to suggest mutations in a remote site of a sugar binding protein which alter its interaction with the substrate. Encouragingly, the predictions, obtained by combining various computational tools, are validated in experiments, again using multitude of techniques. It all converge to non-trivial conclusions about allosteric effects of mutations on substrate binding. The ability to rationally design remote mutations to modulate ligand binding would be very useful to the field. However, while the single example in this paper is interesting, it is not sufficient to ascertain whether the method is reproducible or estimate how accurate or effective it is.

1. Significantly increasing the number of positions tested in the protein. Positions that are conserved (coevolved) as well as unconserved, at a range of distances from the binding site. This would produce a sufficiently large dataset to draw conclusions relating to whether the approach works (n=1 does not allow this) and importantly, why it works.

The authors prioritize evolutionary coupling scores with an ad-hoc approach, considering somehow all the scores of three servers. Otherwise, it is not min DFI and max DCI. Even with slight changes of EV scores, more favorable DFI and DCI cases would have been chosen. Figure 1 shows many residues near I34 (e.g., along the same b-stand and in the b-strand below it) which are equally rigid and coupled. So why was this particular position selected? The text argues that because it also emerged from coevolution analysis, and was over 15 Ang from the active site. However, the decision on a threshold of a distance of 15 Ang from the binding site seems arbitrary. With 10-11 Ang distance, there are amino acid positions with lower DFI and higher DCI values. Also, farther than 15 Ang distance, there are also cases that would be of importance. What is the significance (or somehow confidence levels) of variations in each of these scores (in each of coevolution scores) as well as in DFI and DCI?

Now, let us consider an extreme view: that, from some unknown reason, coevolution provides all the information. For the sake of argument, let us examine the following criteria: high evolutionary coupling scores and distance over 15 Ang. That is, taking the dynamics out of the equation. If it would lead to the same choice then dynamics is not necessarily at play.

2. Greater analysis of the role of solvation water and changes on ligand binding could help rationalize the ITC data. Enthalpy-entropy compensation is common when binding proteins are mutated, previous studies on this should be consulted and referenced.

3. That dynamic coupling underlay allostery is intuitive. However, the link between allostery and evolutionary coupling is not. How would evolutionary coupling between a remote site and the binding site be related to allostery? Mechanistically, evolutionary coupling means that variations in amino acids in one position are compensated by complementary changes in another position. So why would a variation in position I34 be necessarily compensated by variations in the binding site? Why not in other positions? Evolutionary coupling pathways between regulatory sites (e.g., where an effector would bind) and the binding site have been shown, and make sense (evolution 'works' to maintain the required allostery). But why I34? Is it a known regulatory site? What biological function does it facilitate? The authors should at least discuss this. Or, better, examine the possibility of designing mutations without the evolutionary coupling consideration.

4. The choice of the mutations. Table S2 appears to show that the most frequent amino acids in position I34 (in addition to I) are L and F, rather than Y. So how was Y selected?

Further on this, "We also selected K as a mutant, because it is the only positively charged amino acid observed at position 34": That much is true, but why aiming for a positive charge in this position? Perhaps a better argument is that this position features aromatic and hydrophobic residues, making K unique.

5. Binding energy prediction. The -6.0 kcal/mol threshold looks magical. The various cases are assigned scores within a range of ~-6.0 +/- 0.5 kcal/mol. With that, all mutants with binding scores more negative than -6.0 really bind in experiments (dG around -5.5 kcal/mol), while the I34K mutant with a score of -5.85, 0.15 kcal/mol above the threshold, does not bind at all. Not even weakly. The most likely explanation is that the authors were lucky… They should acknowledge that.

It is noteworthy that the binding energies were calculated with the docked poses, presumably before having the crystal structures. However, maybe now, that the crystal structures of the holo and apo protein (and the simulations) are available, it is possible to improve the calculations. Maybe that would give a chance to improve the prediction values? And why having only unbound MD simulations although the authors obtained crystals of the complex structure in unbound and bound states? That is, to carry out MD analysis in both unbound and bound states on both wildtype and mutant (complex AB – (chain A + chain B)) to dissect fully what the story is in the allosteric modulation. By the way, since the authors have the crystal structure of the complex, it makes sense to compare the observed binding poses to the most favorable poses from docking.

6. Entropy gain. To show the entropy gain with the mutation (to interpret the ITC experiments), it should be important to comparatively analyze the unbound and bound wildtype and mutant MD simulations. Is it true that only binding site residues are responsible for this entropy gain? The authors observed changes in the volume of the binding cavity in unbound wildtype vs. mutant structures. But what about changes in the DFI values of binding site residues? If the DFI values are similarly restricted in the unbound and bound states of the mutant structure, and similarly relaxed in bound and unbound states of the wildtype, that would mean that they are not responsible for the entropy gain measured in the experiments.

With allosteric mutations, the redistribution of fluctuations may lead to entropic gain, which would not be fully explained with only considering binding site behavior. The whole structure should be analyzed.

7. Presentation. Title: What is "CV-N"? Maybe other readers also do not know of the top of their heads? Is it possible to avoid using this term in the title? Abstract: "Our results point to a novel approach to identify and substitute distal sites by integrating evolutionary inference with protein dynamics in glycan-binding proteins to improve binding affinity." Overselling. This statement would be legitimate after successful design of, say, 20 mutations in several different proteins. So far there are only 3 mutations in a single position in one protein.

---

## [Author Response]

Essential revisions:1. Significantly increasing the number of positions tested in the protein. Positions that are conserved (coevolved) as well as unconserved, at a range of distances from the binding site. This would produce a sufficiently large dataset to draw conclusions relating to whether the approach works (n=1 does not allow this) and importantly, why it works.The authors prioritize evolutionary coupling scores with an ad-hoc approach, considering somehow all the scores of three servers. Otherwise, it is not min DFI and max DCI. Even with slight changes of EV scores, more favorable DFI and DCI cases would have been chosen. Figure 1 shows many residues near I34 (e.g., along the same b-stand and in the b-strand below it) which are equally rigid and coupled. So why was this particular position selected? The text argues that because it also emerged from coevolution analysis, and was over 15 Ang from the active site. However, the decision on a threshold of a distance of 15 Ang from the binding site seems arbitrary. With 10-11 Ang distance, there are amino acid positions with lower DFI and higher DCI values. Also, farther than 15 Ang distance, there are also cases that would be of importance. What is the significance (or somehow confidence levels) of variations in each of these scores (in each of coevolution scores) as well as in DFI and DCI?Now, let us consider an extreme view: that, from some unknown reason, coevolution provides all the information. For the sake of argument, let us examine the following criteria: high evolutionary coupling scores and distance over 15 Ang. That is, taking the dynamics out of the equation. If it would lead to the same choice then dynamics is not necessarily at play.

As suggested by reviewers, we did significant amount of computational & experimental work, and added more categories to justify our integrated coevolution and dynamic coupling (ICDC) approach. In order to test the ICDC approach, first we applied it to TEM-1 β-lactamase protein, where all possible single point mutations and their impact to ampicillin degradation is available (Stiffler et al., 2015). We calculated the flexibility of each position, and their dynamic couplings and co-evolutionary coupling with the active sites, then created four unique categories in a binary fashion: Category (1,1) dynamically, and coevolutionarily coupled rigid sites; category (1,0) dynamically coupled but co-evolutionarily not coupled sites; category (0,1) dynamically not coupled but co-evolutionarily coupled sites; and category (0,0) dynamically and coevolutionarily not coupled flexible sites. We discuss the details on pages 6 to 9 in the manuscript with addition of a new figure as Figure 1. Briefly, we show that all the single point mutations of category (1,1) positions have the highest impact in both significantly enhancing and reducing the enzymatic activity, irrespective of their position in 3-D structure. In addition, all category (0,0) residue mutations (i.e., the exact opposite of category (1,1)) displays neutral activity. This analysis strongly supports our hypothesis that combining co-evolution and dynamic coupling in identifying putative allosteric distal sites whose mutations could potentially impact function.

We also obtained the positions belonging to each ICDC category for CVN and analyzed how the various substitutions at these identified positions modulate dimannose binding. We modeled a total of 104 variants using MD simulations and predicted dimannose binding scores with our Adaptive BP-Dock docking tool. We generated the distribution of docking energy scores relative to the P51G-m4 binding score for each category added as Figure 2 on page 12.

The details are discussed on pages 10 to 14 in the manuscript.

Briefly, the substitutions of the amino-acid positions in category (1,1) presents a wide range of binding modulation. Similar to TEM-1 results, category (1,1) contains residues that can enhance the binding the most compared to all other categories. Furthermore, the substitutions in category (1,0), when compared to those in category (1,1) share similar favorable binding energy scores, but category (1,1) has a higher number of binding enhancing substitutions. In addition, category (0,0) substitutions show neutral behavior. This also agrees with the same trend obtained with the analysis of TEM-1. We discuss the details on pages 13 and 14 in the manuscript.

Due to the complexity in glycan interactions, it is not possible to design a high throughput experiment for CV-N to compare all mutations that are computationally studied. Thus, we perform two other mutants, A71S/T from category (1,1) in addition to I34K/L/Y to test the proposed mechanism of allosteric modulation of binding pocket dynamics through substitutions (Please see reply 2).

2. Greater analysis of the role of solvation water and changes on ligand binding could help rationalize the ITC data. Enthalpy-entropy compensation is common when binding proteins are mutated, previous studies on this should be consulted and referenced.

We agree additional computational work is needed to establish a solid realization for the entropy-enthalpy compensation. To this extent, MD simulations of P51G-m4 and I34YK/L/Y variants in complex with dimannose have been performed. The simulation trajectories were subjected to Molecular Mechanics Poisson-Boltzmann Surface Area (MM125 PBSA) approach to estimate binding free energies and related enthalpic and entropic contributions (Table S4). The obtained computational free energies of the wild type and the variants under investigation show great promise on capturing respectively correct ranking with experimental results emerged from ITC analysis. The computational entropy values are in alignment with the ITC analysis showing the role of entropy compensation in modulation of binding affinity upon mutations far from the binding site. These results combined with the DFI analyses of complex vs unbound CV-N strengthen the dynamic allosteric effect of distal mutations in modulation of binding affinity. We discuss the details on pages 21 to 28 in the manuscript and added necessary references regarding entropy (Breiten et al., 2013; Chodera and Mobley, 2013; Cornish-Bowden, 2002; Fox et al., 2018).

3. That dynamic coupling underlay allostery is intuitive. However, the link between allostery and evolutionary coupling is not. How would evolutionary coupling between a remote site and the binding site be related to allostery? Mechanistically, evolutionary coupling means that variations in amino acids in one position are compensated by complementary changes in another position. So why would a variation in position I34 be necessarily compensated by variations in the binding site? Why not in other positions? Evolutionary coupling pathways between regulatory sites (e.g., where an effector would bind) and the binding site have been shown, and make sense (evolution 'works' to maintain the required allostery). But why I34? Is it a known regulatory site? What biological function does it facilitate? The authors should at least discuss this. Or, better, examine the possibility of designing mutations without the evolutionary coupling consideration.

In general, coevolution methods fall into two categories (reviewed here (de Juan et al., 2013)). The first method is designed to identify direct contacts (i.e. residue pairs) in tertiary structures which is heavily used in structure prediction (Hopf et al., 2018; Jumper et al., 2021; Morcos et al., 2011; Wang et al., 2016, 2016; Xu, 2019). The approach is based on maximizing the entropy through the whole sequence over MSA to identify global networks of evolutionarily correlated positions governing the fold. The second method seeks detecting correlated amino acid substitutions in pairs of position related with function, this method uses local entropy (i.e. Mutual information, statistical coupling analysis (Lockless and Ranganathan, 1999)). It has been shown that this type of local statistical information could map networks of coevolving residues that support protein function and has been used to find allosteric networks (Salinas and Ranganathan, 2018, Reynolds et al., 2011). The two methods sample different parts of the information content within protein sequences and uncover different elements of protein structure, thus they complement each other. So, we wanted to use both to capture sites coevolved with binding site while maintaining fold and function. We have added these citations and revised manuscript accordingly.

The goal of our ICDC approach is to computationally predict the possible allosteric (aka distal) mutations that can finetune binding affinity. As both co-evolution and coupling dynamics use different inputs to infer correlations albeit having some noise, we hypothesize that combination of these two approaches could provide putative allosteric sites. As discussed in response to comment 1, we expanded our hypothesis by creating multiple categories and to test it applied it on TEM-1 protein system where functional data regarding all possible single point mutations and how they alter activity is available. When ICDC is applied on CV-N, we found I34 as the most promising candidate and the strongest position in our prediction to modulate binding, thus we experimentally characterize it in detail initially and then extended our experimental and computational work on A71 location. The details of ICDC approach on CVN and the results are on pages 10 to 26 in the manuscript.

Furthermore, we selected two more variants A71S/T from category (1,1), to further test the hypothesis. The A71T variant displays binding altering affects similar to I34Y variant. Both the rigidification of binding sites, the binding energy predictions with X-score energy unit (i.e., XEU), and MM-PBSA free energy estimations upon A71T mutation resembles the I34Y variant. Moreover, the A71S variant is found to be showing a dynamic behavior close to I34L with our computational results. To understand these changes better we experimentally characterized both A71T and A71S mutations. The ITC results agree with the computational findings, importantly, the trend previously discussed with I34Y entropy/enthalpy compensation mechanism can be seen with A71T variant as well. The results observed with an additional category (1,1) position further confirms the distinction in binding modulation created by combining dynamics and coevolution with our ICDC approach. The details of these results are discussed in pages 26 to 28.

4. The choice of the mutations. Table S2 appears to show that the most frequent amino acids in position I34 (in addition to I) are L and F, rather than Y. So how was Y selected?Further on this, "We also selected K as a mutant, because it is the only positively charged amino acid observed at position 34": That much is true, but why aiming for a positive charge in this position? Perhaps a better argument is that this position features aromatic and hydrophobic residues, making K unique.

We thank the reviewer for the comment, and we agree the text was not clear on our selection of amino acids on position I34. We revised the text accordingly and included all the available amino acids from the MSA in our calculations and analyses. While we computationally performed all possible substitutions based on MSA, three of them where we predict diverse functional outcomes (i.e., abolishing binding, neutral and enhancement in binding) are subjected to experimental characterization. We discuss the details on pages 14 to 18 in the manuscript.

5. Binding energy prediction. The -6.0 kcal/mol threshold looks magical. The various cases are assigned scores within a range of ~-6.0 +/- 0.5 kcal/mol. With that, all mutants with binding scores more negative than -6.0 really bind in experiments (dG around -5.5 kcal/mol), while the I34K mutant with a score of -5.85, 0.15 kcal/mol above the threshold, does not bind at all. Not even weakly. The most likely explanation is that the authors were lucky… They should acknowledge that.

The binding energies are calculated using X-score (Wang et al., 2002) in our Adaptive BP-Dock software (Bolia and Ozkan, 2016). Thus, binding energies are not actual kcal/mol or in particular 1 kcal/mol of experimental binding free energy does not correspond to 1 kcal/mol Xscore energy. To make this clear, we change the energy score units to X-score Energy Unit (XUE). In that X-score energy unit, we have tested different proteins (including CV-N) (Bolia et al., 2014; Li et al., 2015; Woodrum et al., 2013) and -6.0 XEU is indeed critical and we showed 0.5 XEU makes a difference in binding affinity (See table 1 from Bolia et al., 2014b) when we studied single point mutations of CV-N in our earlier work.

It is noteworthy that the binding energies were calculated with the docked poses, presumably before having the crystal structures. However, maybe now, that the crystal structures of the holo and apo protein (and the simulations) are available, it is possible to improve the calculations. Maybe that would give a chance to improve the prediction values? And why having only unbound MD simulations although the authors obtained crystals of the complex structure in unbound and bound states? That is, to carry out MD analysis in both unbound and bound states on both wildtype and mutant (complex AB – (chain A + chain B)) to dissect fully what the story is in the allosteric modulation. By the way, since the authors have the crystal structure of the complex, it makes sense to compare the observed binding poses to the most favorable poses from docking.

We thank the reviewer for the suggestions. We carried out MD simulations in bound states in addition to the unbound states. The results of the new MD simulation and old ones with a diverse selection is rewritten under Results section on pages 18 to 28. We additionally compared the dock pose and crystal structure of the bound pose I34Y (Figure 4 —figure supplement 5). The docked pose displays a conformation close to the bound crystal structure.

6. Entropy gain. To show the entropy gain with the mutation (to interpret the ITC experiments), it should be important to comparatively analyze the unbound and bound wildtype and mutant MD simulations. Is it true that only binding site residues are responsible for this entropy gain? The authors observed changes in the volume of the binding cavity in unbound wildtype vs. mutant structures. But what about changes in the DFI values of binding site residues? If the DFI values are similarly restricted in the unbound and bound states of the mutant structure, and similarly relaxed in bound and unbound states of the wildtype, that would mean that they are not responsible for the entropy gain measured in the experiments.With allosteric mutations, the redistribution of fluctuations may lead to entropic gain, which would not be fully explained with only considering binding site behavior. The whole structure should be analyzed.

As suggested, we employed MD simulations on both unbound and bound states of the variants. We obtained the DFI profiles of the binding site residues using both bound and unbound simulations of the wildtype and the variants. The details are on pages 22 to 28 in the manuscript. Briefly, when we compare the change in DFI profiles of the binding residues of the variant I34Y with respect to wild type, we observe that positions T57 is much more rigid compared to wild type for unbound dynamics. However, position T57 exhibits a similar rigidity when the bound simulations are considered. This analysis suggests that rigidification of T57 in the unbound form of the I34Y variant plays a critical role in entropic compensation upon dimannose binding. This trend in change in flexibility also agrees with the entropy change found with MM-PBSA approach and ITC results. We revised the text and added additional bound flexibility analysis as suggested.

Additionally, in order to understand to what extend a variation in a category (1,1) positions altered the binding sites dynamics and brings out a compensation between entropy and enthalpy, we investigated the total change in flexibility (change in %DFI) vs ΔG of binding from experiments of binding sites (Figure 7 —figure supplement 1). First, when the binding enhancing variants were inspected, we observed a rigidification in the binding region upon mutation. On the other hand, in the binding abolishing variant (e.g., I34K) the binding site residues are much more flexible compared to the wild type. Moreover, we selected residues highly dynamically coupled to binding sites and investigated their change in flexibilities to understand their importance in binding modulation. The total change in flexibility of these residues agrees with the ΔG of binding similar to binding residues themselves. Further, as a control, we randomly selected residues and calculated their change in flexibilities and observed poor correlations with ΔG of binding. Ultimately, we show that the total change in flexibility of binding sites and the residues highly coupled to them correlates well with the experimentally measured ΔG of binding by ITC. The details of this part of the study are discussed on pages 22 to 28.

7. Presentation. Title: What is "CV-N"? Maybe other readers also do not know of the top of their heads? Is it possible to avoid using this term in the title? Abstract: "Our results point to a novel approach to identify and substitute distal sites by integrating evolutionary inference with protein dynamics in glycan-binding proteins to improve binding affinity." Overselling. This statement would be legitimate after successful design of, say, 20 mutations in several different proteins. So far there are only 3 mutations in a single position in one protein.

We use Cyanovirin-N in title, and revised the text as the results point out our integrated coevolution and dynamic coupling (ICDC) approach can identify distal residues and find substitutions on these sites which can modulate activity for enzymes and glycan binding for lectins.

References

Bolia A, Ozkan SB. 2016. Adaptive BP-Dock: An Induced Fit Docking Approach for Full Receptor Flexibility. *J Chem Inf Model* 56:734–746. doi:10.1021/acs.jcim.5b00587

Bolia A, Woodrum BW, Cereda A, Ruben MA, Wang X, Ozkan SB, Ghirlanda G. 2014. A Flexible Docking Scheme Efficiently Captures the Energetics of Glycan-Cyanovirin Binding. *Biophys J* 106:1142–1151. doi:10.1016/j.bpj.2014.01.040

Breiten B, Lockett MR, Sherman W, Fujita S, Al-Sayah M, Lange H, Bowers CM, Heroux A, Krilov G, Whitesides GM. 2013. Water Networks Contribute to Enthalpy/Entropy Compensation in Protein–Ligand Binding. *J Am Chem Soc* 135:15579–15584. doi:10.1021/ja4075776

Campitelli P, Swint-Kruse L, Ozkan SB. 2021. Substitutions at Nonconserved Rheostat Positions Modulate Function by Rewiring Long-Range, Dynamic Interactions. *Mol Biol Evol* 38:201–214. doi:10.1093/molbev/msaa202

Chodera JD, Mobley DL. 2013. Entropy-enthalpy compensation: Role and ramifications in biomolecular ligand recognition and design. *Annu Rev Biophys* 42:121–142. doi:10.1146/annurev-biophys-083012-130318

Cornish-Bowden A. 2002. Enthalpy—entropy compensation: a phantom phenomenon. *J Biosci* 27:121–126.

de Juan D, Pazos F, Valencia A. 2013. Emerging methods in protein co-evolution. *Nat Rev Genet* 14:249–261. doi:10.1038/nrg3414

Fox JM, Zhao M, Fink MJ, Kang K, Whitesides GM. 2018. The Molecular Origin of Enthalpy/Entropy Compensation in Biomolecular Recognition. *Annu Rev Biophys* 47:223–250. doi:10.1146/annurev-biophys-070816-033743

Hopf TA, Schärfe CPI, Rodrigues JPGLM, Green AG, Kohlbacher O, Sander C, Bonvin AMJJ, Marks DS. 2018. Sequence co-evolution gives 3D contacts and structures of protein complexes. *eLife* 3. doi:10.7554/*eLife*.03430

Jumper J, Evans R, Pritzel A, Green T, Figurnov M, Ronneberger O, Tunyasuvunakool K, Bates R, Žídek A, Potapenko A, Bridgland A, Meyer C, Kohl SAA, Ballard AJ, Cowie A, RomeraParedes B, Nikolov S, Jain R, Adler J, Back T, Petersen S, Reiman D, Clancy E, Zielinski M,

Steinegger M, Pacholska M, Berghammer T, Bodenstein S, Silver D, Vinyals O, Senior AW, Kavukcuoglu K, Kohli P, Hassabis D. 2021. Highly accurate protein structure prediction with AlphaFold. *Nature* 596:583–589. doi:10.1038/s41586-021-03819-2

Li Z, Bolia A, Maxwell JD, Bobkov AA, Ghirlanda G, Ozkan SB, Margulis CJ. 2015. A Rigid Hinge Region Is Necessary for High-Affinity Binding of Dimannose to Cyanovirin and Associated Constructs. *Biochemistry* 54:6951–6960. doi:10.1021/acs.biochem.5b00635

Lockless SW, Ranganathan R. 1999. Evolutionarily conserved pathways of energetic connectivity in protein families. *Science* 286:295–299.

Morcos F, Pagnani A, Lunt B, Bertolino A, Marks DS, Sander C, Zecchina R, Onuchic JN, Hwa T, Weigt M. 2011. Direct-coupling analysis of residue coevolution captures native contacts across many protein families. *Proc Natl Acad Sci* 108:E1293–E1301. doi:10.1073/pnas.1111471108

Stiffler MA, Hekstra DR, Ranganathan R. 2015. Evolvability as a Function of Purifying Selection in TEM-1 β-Lactamase. *Cell* 160:882–892. doi:10.1016/j.cell.2015.01.035

Wang R, Lai L, Wang S. 2002. Further development and validation of empirical scoring functions for structure-based binding affinity prediction. *J Comput Aided Mol Des* 16:11–26. doi:10.1023/A:1016357811882

Wang S, Li W, Zhang R, Liu S, Xu J. 2016. CoinFold: a web server for protein contact prediction and contact-assisted protein folding. *Nucleic Acids Res* 44:W361–W366. doi:10.1093/nar/gkw307

Woodrum BW, Maxwell JD, Bolia A, Ozkan SB, Ghirlanda G. 2013. The antiviral lectin cyanovirinN: probing multivalency and glycan recognition through experimental and computational approaches. *Biochem Soc Trans* 41:1170–1176. doi:10.1042/BST20130154

Xu J. 2019. Distance-based protein folding powered by deep learning. *Proc Natl Acad Sci* 116:16856–16865. doi:10.1073/pnas.1821309116